# Mental search of concepts is supported by egocentric vector representations and restructured grid maps

Simone Viganò [1,2] ✉, Rena Bayramova[1], Christian F. Doeller [1,3,4,5] & Roberto Bottini [2,5]

The human hippocampal-entorhinal system is known to represent both spatial locations and abstract concepts in memory in the form of allocentric cognitive maps. Using fMRI, we show that the human parietal cortex evokes complementary egocentric representations in conceptual spaces during goal-directed mental search, akin to those observable during physical navigation to determine where a goal is located relative to oneself (e.g., to our left or to our right). Concurrently, the strength of the grid-like signal, a neural signature of allocentric cognitive maps in entorhinal, prefrontal, and parietal cortices, is modulated as a function of goal proximity in conceptual space. These brain mechanisms might support flexible and parallel readout of where target conceptual information is stored in memory, capitalizing on complementary reference frames.

A crucial characteristic of an intelligent mind is the ability to store and manipulate the knowledge it has about the world, and to organize concepts and memories for later retrieval. Cognitive (neuro) scientists have proposed that humans organize their world knowledge in "cognitive maps"[1,2], internal models of how memories relate to each other[3–6]. The hippocampal–entorhinal system in the medial temporal lobe of mammals contains spatially modulated neurons, such as place cells[7] or grid cells[8] that, by selectively firing when the animal is at a specific location in the environment, provide a neural basis for cognitive maps of their surroundings. Functional magnetic resonance imaging (fMRI) in humans has revealed that similar coding schemes in the same brain regions and in the medial prefrontal cortex can be detected non-invasively while participants are engaged in virtual reality navigation tasks[9–11] and also when, in more abstract decision-making, they need to process relational knowledge between non-spatial stimuli, such as visual shapes or objects[12,13], odors[14], people's identities[15], or word meanings[16,17]. This supports the proposal of a phylogenetic continuity between spatial navigation abilities and memory in humans[18]: the brain circuits that evolved for

navigating physical spaces in other mammals might be used in our species for organizing conceptual knowledge, enabling us to "mentally navigate" through concepts and memories as if they were locations in our internal conceptual spaces[3,6,19] (but see also ref. [20] for potential evidence in rodents). However, the navigation system goes well beyond hippocampal cognitive maps. During physical navigation in real and virtual environments, wayfinding and spatial self-localization is performed by means of the parallel recruitment of complementary reference frames: *allocentric* representations (indeed associated with the hippocampal formation) encode objects' position in relationship with other objects, stable landmarks or environmental geometry; on the other hand, *egocentric* representations (usually associated with parietal regions) encode object position with respect to the observer's perspective, and changes as a function of the observer's movements[21–23] (for similar findings in non-human species see refs. [24–27]). It has been hypothesized that complementary allocentric and egocentric reference frames may also be recruited during the exploration of conceptual spaces[6], suggesting a repurposing of the wider human navigation network (hippocampal-

[1]Max Planck Institute for Human Cognitive and Brain Sciences, Leipzig, Germany. [2]Center for Mind/Brain Sciences, University of Trento, Rovereto, Italy. [3]Kavli Institute for Systems Neuroscience, Centre for Neural Computation, The Egil and Pauline Braathen and Fred Kavli Centre for Cortical Microcircuits, Jebsen Centre for Alzheimer's Disease, Norwegian University of Science and Technology, Trondheim, Norway. [4]Wilhelm Wundt Institute of Psychology, Leipzig University, Leipzig, Germany. [5]These authors jointly supervised this work: Christian F. Doeller, Roberto Bottini. ✉e-mail: vigano@cbs.mpg.de

parietal system) for conceptual 'navigation', but empirical evidence to date is still missing.

Here we test this hypothesis by investigating how the human brain represents and guides the retrieval of relevant goal information in conceptual spaces during mental search. Indeed, when looking for specific goal objects or locations during physical navigation, both egocentric and allocentric mechanisms guide our search in the environment. First, egocentric or viewpoint-dependent representations allow us to tell whether an object is, at a specific moment in time, to our left, right, front, or back, thus providing information about its location relative to our current position or movement trajectory (that is, our current state). The neural circuits supporting this egocentric coding rely mainly on the parietal cortex, as confirmed by neuropsychology[28-30], functional neuroimaging[31-34], and neurophysiology[35,36]. Second, goals can also be tracked by altering and restructuring the allocentric cognitive maps in the hippocampal formation: goal locations are indeed overrepresented by spatially tuned neurons compared to more distant regions of space[37], as it happens for instance with entorhinal grid cell fields, that expand and shift towards goals[38] and are more densely packed at the goal location[38,39]. Is it possible that specific goal

information in conceptual spaces elicits similar neural effects across different reference frames? Do they evoke egocentric-like coding schemes when we look for them in memory, to orient and guide our mental search? Do they alter the allocentric cognitive map to signal their position in conceptual space?

To answer these questions we designed an experiment, inspired by previous work[12], where human participants had to retrieve conceptual goals in simple, but highly controlled feature spaces, allowing us to monitor their brain activity as a function of their "mental search". We instructed 40 adult volunteers to imagine being scientists in a future society where humankind lives in peace and prosperity thanks to the development of two (fictitious) molecules, called *Megamind* and *Megabody*, which grant people increased cognitive and physical abilities, respectively. The two molecules that represented the goals were identical for all but two crucial aspects: their color (blue vs green, hereafter referred to as "contexts"), and the specific ratios of their upper and lower bonds (hereafter referred to as "bond-length ratio"), see Fig. 1a. Unbeknown to participants, the two molecules were defined as points in 2D feature spaces, with one axis indicating the length of the upper bond and the other one the length of the lower

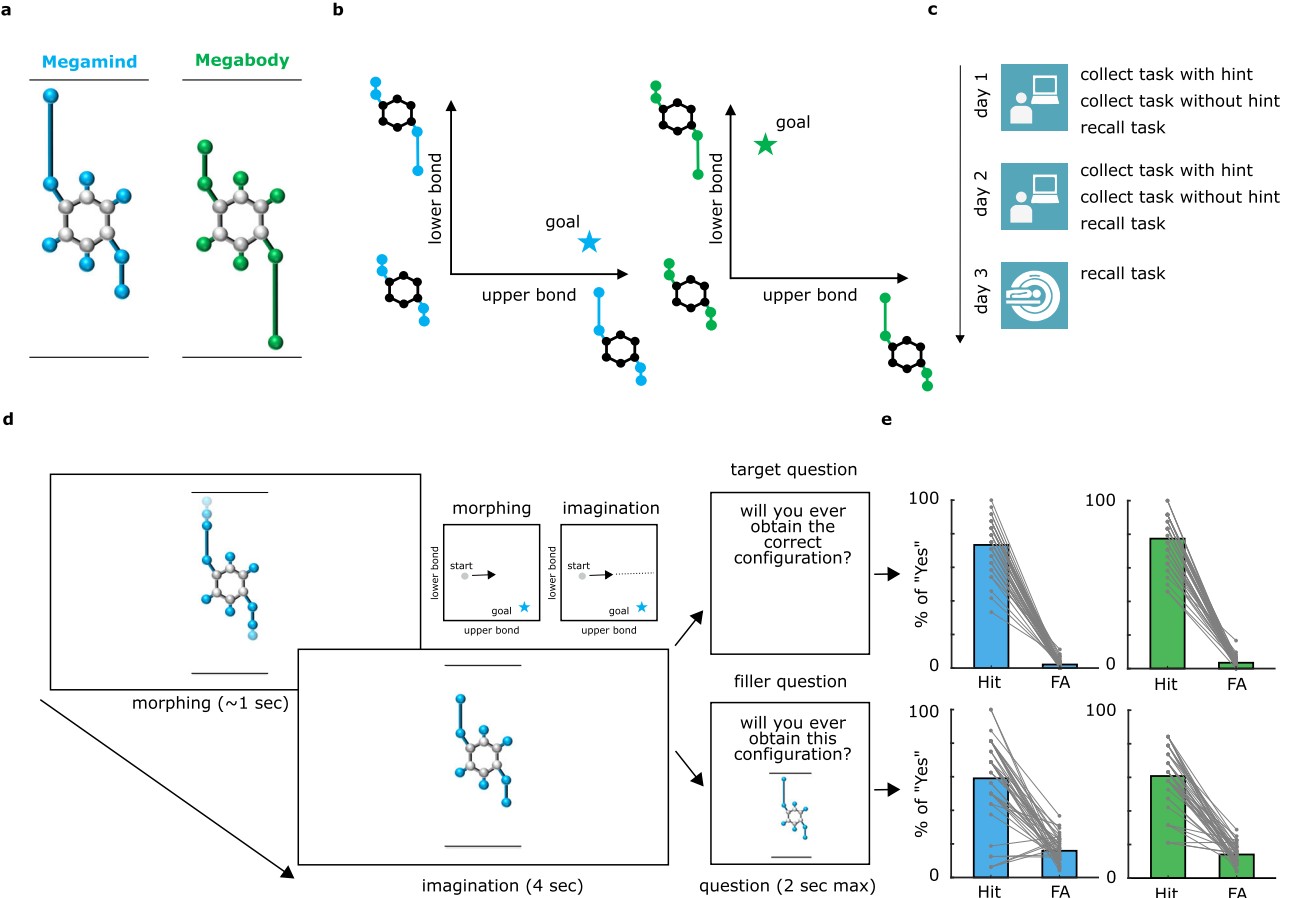

**Fig. 1 | Conceptual goals in feature space and experimental design.**
**a** Participants learned about two fictitious goal molecules, Megamind and Megabody, with their characteristic colors and ratios between the upper and lower bond ("bond-length ratio"). **b** The two-goal molecules were conceivable as goal locations in bidimensional feature spaces defined by the length of the two bonds. **c** The experiment lasted 3 days. During the first 2 days, participants learned how to "produce" the goal molecules by actively morphing feature configurations into the target ones (thus effectively 'navigating' the feature spaces to find goal locations), via computer-based behavioral training. Further details on these training tasks and their results are in the Methods section and in Supplementary Figs. 1–3. On the third and last day, during an fMRI scanning session, participants evaluated some reactions while keeping in mind conceptual goals. **d** The "Recall task" was performed

during training and during fMRI. Participants were prompted with a random feature configuration which started morphing autonomously for about 1 s. After the morphing stopped, they had 4 s to imagine the morphing continuing in the same way. Then, one of two possible questions appeared. The "Target question" asked whether the goal molecule would ever be obtained if the morphing continued in the same way. The "Filler question" asked whether a specific molecule configuration (different from the goal molecule) shown on the screen would ever be obtained. Please refer to the Supplementary Movies for examples of the tasks. **e** The bar plots show the percentage of hits and false alarms for the two molecules and the two questions on the fMRI day across the $n = 40$ participants. Performance on the training days is shown in Supplementary Figs. 4–7.

bond (Fig. 1b), see Methods for further details. Participants were told that they were among the scientists recruited to produce these molecules via specific computer-based tasks over the course of 2 days (Fig. 1c). To produce the goal molecules, they had to actively morph feature configurations into the target ones using computer-controlled "reactions" (see Methods and Supplementary Movie 1), and this was conceivable as navigating the two "molecule spaces" (or contexts) in order to find the conceptually relevant labeled location (our "conceptual goals"), Supplementary Fig. 1a. Then, during a subsequent fMRI session, participants were presented with predetermined reactions, showing morphing molecules whose bond-length ratio changed in a predetermined way (that is, following specific trajectories) (see Fig. 1d and Supplementary Movie 2). They were asked to imagine the morphing to continue in the same way and to indicate whether or not the reaction would have resulted in the molecule becoming Megamind or Megabody. We predicted that, analogous to spatial navigation, brain activity during abstract conceptual search would convey information about goal locations in feature spaces by exhibiting (1) egocentric-like coding of the conceptual goals, and (2) an alteration of the allocentric cognitive map as a function of goal proximity.

## Results

### Participants successfully retrieve conceptual goals in feature spaces

Participants were trained for two consecutive days to find the goal molecules in the two contexts: they learned to actively morph wrong molecules into correct ones by modifying their bond–length ratio to match either Megamind or Megabody depending on the color context (see Methods and Supplementary Fig. 1). Unbeknown to participants, this was conceivable as navigating the molecule space to find goal locations. Participants successfully completed the training, as indicated by the time to finish the tasks, the number of transitions necessary to find the goals, and the number of correct molecules collected in a given time window (Supplementary Figs. 1 and 2), suggesting that they had learned and memorized how the conceptual goals looked like or, equivalently, where the conceptual goals were located in the two contexts. At the end of each training day and during the fMRI session (day 3), participants performed the "Recall task" (see Methods), consisting of the presentation of a random morphing molecule and of an imagination period (Fig. 1d). Here, participants were instructed to imagine the morphing ("reaction") to continue infinitely in the very same way, and to imagine whether or not the molecule would ever match the goal configuration (that is, in spatial terms, if the trajectory would ever pass through the goal location), which they had learned during training and which they had to keep in mind. This was referred to as the "target question" (Fig. 1d). Critically, the trajectories implied by the morphing molecules could be directed to the goal (0° from correct trajectory; on-target trials, participants are expected to respond "Yes") or deviating from it by a certain degree (−135°, −90°, −45°, +45°, +90° +135° from correct trajectory; off-target trials, the expected response was "No"). Signs could be interpreted as arbitrary "left" and "right", see Methods and Supplementary Fig. 3): these angular conditions will be referred to as "egocentric-like conditions", because they indicate where the goal is in the feature space with respect to the current trajectory, independently from their allocentric position and from the angle of movement relative to the general layout of the feature space (Fig. 2a). In other words, as it happens in the physical environment where, given an oriented agent ("viewpoint") positioned in it, and a specific goal or location to refer to (the "target"), we can define whether the target is positioned to one side ("left") or the other ("right") of our current point of view, here we had an environment (the molecule space), a target (the goal molecule), and a point of view (defined by the morphing direction of the molecule) that allowed us to mimic the "one side vs the other side" (left vs. right) distinction.

Moreover, to ascertain that participants were also actively constructing a representation of the entire navigable space, on 50% of the trials, instead of the "target question", we presented a "filler question", where we asked participants to detect whether or not the morphing would have resulted in a specific, randomly chosen, configuration that was different from the goal and that was presented on the screen only for that specific trial (see Methods). Participants did not know in advance the type of question they were going to be presented with and, by analyzing the "imagination period" in our fMRI data, we could address all the main questions of our study without significant influence from the "question period". Participants had a very good performance on both training days: they successfully distinguished correct trajectories from incorrect ones in both the "target questions" (Supplementary Fig. 4) and the "filler questions" (Supplementary Fig. 6). The analysis of reaction times (RTs) showed only modest differences between correct and incorrect trajectories, mostly in the "filler questions", with on-target trials being recognized slightly more slowly than off-target ones (Supplementary Figs. 5 and 7). This task was repeated on the 3rd and last day of the experiment, during the fMRI scanning session (see Methods), where participants maintained their high performance (target question: 73.3% of hits and 2.11% of false alarms for the blue molecule, 77.4% of hits and 3.47% of false alarms for the green molecule; filler question: 59% of hits and 15.9% of false alarms for the blue molecule, 60.8% of hits and 14% of false alarms for the green molecule, Fig. 1d). There was no statistically significant difference in behavioral performance between the two molecules (all ps > 0.10), nor between the trials that implied different off-target conditions (see Methods and Supplementary Figs. 4–7).

### Egocentric-like representations of conceptual spaces in medial parietal cortex: evidence from fMRI adaptation

To examine whether conceptual goals elicited egocentric-like codes during mental search, we applied an fMRI adaptation analysis previously used to investigate brain activity during allocentric spatial and conceptual navigation[9,17]: we looked for brain regions where activation was reduced as a function of the time passed since the same egocentric-like condition (e.g., direction of 45° right to the goal) was presented (Fig. 2b). Whole-brain analysis revealed significant adaptation in the medial occipito-parietal cortex, extending along the parieto-occipital sulcus and the precuneus (peaks at $MNI_{x,y,z}$ coordinates: 6 −92 22, −12 80 48, and −4 −76 30, $t$ = 5.83, 5.68, 5.67 respectively, Fig. 2c), similar to previous studies investigating egocentric reference frames in spatial tasks[31–34]. Smaller clusters were observed in the superior frontal gyrus, close to frontal eye fields ($MNI_{x,y,z}$: −28 28 54, $t$ = 6.89), and in other visual and parietal regions, see Supplementary Table 1. Notably, the adaptation effect in the medial occipito-parietal cortex positively correlated with participants' performance (Pearson's correlation with % of hit; $r$ = 0.37, $p$ = 0.0180, see Fig. 2d), and it was present when we excluded on-target trials (Fig. 2e "off-target" and Supplementary Fig. 8), when we focused on each molecule separately (Fig. 2e "blue" and "green" and Supplementary Fig. 9), and when we controlled for distance from the goal (Fig. 2e "distance controlled" and Supplementary Fig. 10, showing main effect in the hippocampus). The effect was also present (although weaker) when we repeated the analysis only considering the left vs. right position of the goal ignoring the angular distance (Fig. 2e "left vs. right" and Supplementary Fig. 11), when we controlled for repetition of the starting point (Fig. 2e "starting point controlled" and Supplementary Fig. 12), or when we controlled for the signal representing allocentric "head-direction like" activity (Supplementary Fig. 13).

### Egocentric-like representations of conceptual spaces in medial parietal cortex: evidence from MVPA

To further validate our results, we implemented a whole-brain searchlight multivariate analysis, training a classifier to distinguish

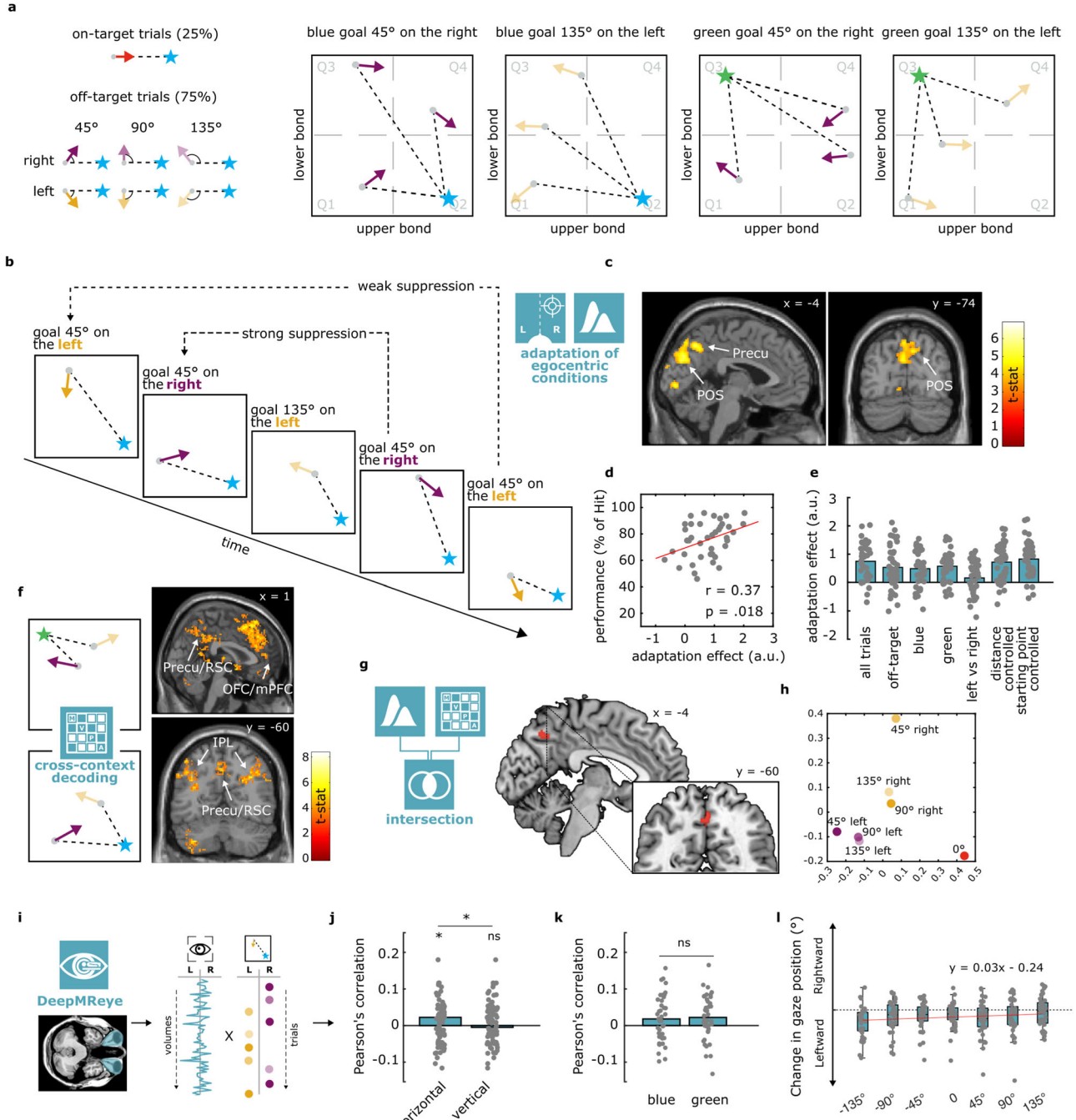

**Fig. 2 | Evidence of egocentric-like coding in the medial parietal cortex.**
**a** Morphing trajectories subtending only 7 possible angular conditions to the goal ("egocentric conditions"). **b** Adaptation approach, modeling fMRI BOLD signal as a function of the log time passed between two presentations of the same egocentric condition. **c** Results of the adaptation analyses, one-sided $t$-test as implemented in SPM12, thresholded at $p < 0.001$ at the voxel level, corrected for multiple comparisons at cluster level with $p < 0.05$. **d** Pearson's correlation (two-sided) between adaptation effect in the medial occipito-parietal cluster and percentage of Hit during the recall task. **e** Control analyses in occipito-parietal cluster, considering (i) only "off-target" trials, (ii) "blue" and (iii) "green" contexts separately, (iv) whether the goal was on the "left vs. right" without considering the angular distance, and egocentric-like responses after controlling for (vi) distance from the goal or (vii) repetition of the same starting point. For all the control analyses mentioned above, Supplementary Figs. 8–12 report the unbiased whole-brain statistical analyses, while this panel reports participants' scores and their average in the main occipito-temporal cluster without statistics, to avoid circularity. $n = 40$ participants.

a.u. = arbitrary units. **f** Cross-context decoding approach, where a classifier is trained to distinguish egocentric conditions in one context and is then tested in the other one, and vice versa (see Methods). Results of a one-sided $t$-test (SPM12) are thresholded at $p < 0.001$ at voxel level, corrected for multiple comparisons at cluster level with $p < 0.05$. **g** Anatomical intersection between the brain maps resulting from the adaptation analysis and the cross-context decoding. **h** MDS of the egocentric conditions. Patterns of activity are extracted from the cluster in the precuneus (peak from intersection analysis). **i** We used the DeepMReye toolbox[41] to estimate gaze behavior from the eyeballs' BOLD signal. Trial-wise eye movements were correlated with the "true" position of the goal in conceptual space. **j** Fisher-transformed Pearson's correlation score was significant at the group level ($n = 40$, two-sided $t$-test) for horizontal eye movements. **k** The effect was present in both contexts (two-tailed $t$-test), although to a weaker extent, without differences between the two. **l** Regression line showing the linear relationship between ego-centric angles of goal position in conceptual space and leftwards vs. rightwards eye movements ($n = 40$). *$p < 0.05$; ns not significantly different.

between egocentric-like angular conditions in one context (e.g., blue molecule) and testing it in the other one (e.g., green molecule, see Methods and Fig. 2e). This represents a complement to univariate adaptation analyses because it focuses on distributed activity across voxels rather than single-voxel changes in BOLD signal (see Methods[40]). This analysis revealed a large and widespread network of brain regions consisting of the OFC/PFC ($MNI_{x,y,z} = -36\ 20\ -6$, $t = 8.29$), the right IPL ($MNI_{x,y,z} = 44\ -40\ 54$ $t = 7.2$), the basal ganglia (BG, $MNI_{x,y,z} = -2\ 4\ -4$, $t = 6.86$), the left middle temporal gyrus (MTG, $MNI_{x,y,z} = -54\ -48\ -10$, $t = 6.72$), the left IPL ($MNI_{x,y,z} = -54\ -32\ 44$, $t = 6.47$), and the posterior parietal cortex (PPC) extending to the retrosplenial cortex (RSC) ($MNI_{x,y,z} = 4\ -40\ 36$, $t = 6.24$) that similarly represented egocentric-like conditions across the two contexts (Fig. 2f). For the PPC-RSC, the effect was significant also for individual contexts ($p = 0.007$ for both blue and green contexts), although weaker compared to the cross-context situation (mean accuracy = 3.93% and 3.84 above chance level for blue and green context separately, while it reached 5.49% above chance level in the cross-context analysis, see also Supplementary Fig. 14). Crucially, the PPC-RSC effect remained significant also when on-target trials were removed (Supplementary Fig. 15). By intersecting the whole-brain statistical maps obtained from the two approaches (univariate adaptation and multivariate decoding, Fig. 2g), we obtained a single cluster in the precuneus, with a center of mass at $MNI_{x,y,z}$ coordinates −1, −59, 43 and an extension of 102 voxel, that was anatomically remarkably close to the brain region reported by Chadwick and colleagues[32] for the representation of egocentric goal directions (front vs. left vs. right vs back) during a spatial orientation task (peak at $MNI_{x,y,z}$ −6 −61 39). We used our resulting cluster to recover a low dimensional projection of how egocentric-like conditions were represented in this region using multidimensional scaling (Fig. 2h), illustrating both the distinct clusters for left vs. right as well as the angular direction-specific representations.

### Egocentric-like representations of conceptual spaces in behavior: evidence from eye movements

To determine the egocentric nature of these representations, we looked for a confirming signature in participants' behavior. In orientation tasks, a typical signature of spatial allocation of attention is provided by eye movements and gaze behavior. We reasoned that if participants conceived the mentally navigated conceptual spaces in egocentric terms, this might be reflected in their spontaneous eye movements. More specifically, we asked whether participants moved their eye to the right or left when the goal was to the "right" or "left" (as in, two opposite sides) in the conceptual spaces, compared to their current stimulation. Rather than being a confounding variable, this would provide more supporting evidence of the true egocentric nature of the representations we reported.

To answer our question we used the recently developed DeepM-REye toolbox[41], a convolutional neural network that decodes gaze position from the magnetic resonance signal of the eyeballs. We used a set of pre-trained weights, freely available from the toolbox website and resulting from previous experiments where concurrent BOLD and eye-tracking signals were recorded[42–44], to obtain the estimates of gaze position for each fMRI volume, in each run and subject. First, we wanted to make sure that the tool could be reliably applied to our dataset. To do so, we focused on the period of the Target question, where participants had to choose between the answers "yes" and "no", which were presented on the left or right side of the screen (counterbalanced across trials). We correlated the trial-by-trial choice of the subjects (selected option on the left or option on the right) with the trial-by-trial gaze change after the question onset, under the assumption that participants are likely to look towards the chosen option. Specifically, we expected a positive correlation with gaze change on the horizontal axis and not with the vertical one, which was confirmed (mean correlation for the horizontal movements = 0.028, $p = 0.0004$).

Then, having demonstrated that reliable left vs. right information can be decoded using deepMReye, we applied the same logic to our question of interest. Specifically, we looked at eye movements indicated by the difference between the starting of the morphing period and the starting of the imagination period: by subtracting the former to the latter, we obtained a positive result if the subjects moved their eyes to the right and a negative one if they moved them to the left (Fig. 2i). We then extracted, for each trial, the implied position of the goal in the conceptual space respective to the current trajectory (that is, the egocentric condition: −135°, −90°, −45°, 0°, +45°, +90°, +135°). We then correlated the trial-by-trial vector of eye movements with the trial-by-trial vector of where the goal should be in the conceptual space from an egocentric point of view. In line with our hypothesis and the above-reported results, we observed a significant positive correlation with the horizontal eye movements only (mean Pearson's correlation = 0.02; $p = 0.022$; vertical eye movements: mean Pearson's correlation = −0.005; $p = 0.53$) (Fig. 2j–l). The effect was present when we focused on both contexts separately, although to a weaker extent due to the smaller number of trials, without a statistically significant difference between them ($p = 0.68$) (Fig. 2k). Taken together these findings indicate that participants were representing the conceptual spaces using egocentric codes and perspectives.

### Evidence of mental realignment of conceptual spaces in the parietal cortex

Egocentric perspective during goal-directed spatial orientation has been linked to the ability to transform and rotate existing memory representations of our surroundings, enabling the retrieval of experienced viewpoints to the goal and correct heading directions[21,45]. A candidate faculty supporting this mechanism is mental rotation[46], namely the ability to realign the representations of objects or scenes held in mind to match their visual appearances. Both neuroimaging and non-invasive brain stimulation indeed indicate that the crucial cortical nodes supporting mental rotation in the human brain are located in the parietal cortex[47,48], consistent with the proposal of a dedicated neuronal population in this region, referred to as "parietal window"[22], that is recruited for egocentric transformations during mental exploration and spatial orientation. We reasoned that a similar mechanism of alignment could also operate during goal-directed navigation in conceptual spaces.

If this hypothesis is correct, a specific prediction follows: the representations of the two contexts or environments (blue vs green molecule) should be aligned, in such a way that the goal quadrant from one conceptual space should be more similar to the goal quadrant in the other conceptual space and, crucially, the non-goal quadrants from the two environments would also correspond across contexts following their relative position to the goal. As an example, consider an observer at the center of the green environment facing the goal quadrant (Q3) in Fig. 3a. According to their point of view, Q1 would be to the left of the goal, and Q4 would be to the right. Now consider the same observer in the blue context, facing the context-relevant goal quadrant (Q2). In this case, Q4 would be to the left, and Q1 to the right. If participants are maintaining the focus of attention from their own current state to the goal state, then we might assume that not only Q3 and Q2 in the green and blue context respectively should be represented similarly (they are the goal quadrants "in front of the observer"), but also that the rest of the quadrants would be represented in the same relationship to each other across the two contexts. This can be seen as an example of "alignment" along the common left-right axis, which is orthogonal to the sagittal plane created by connecting our current state with the goal state and actually dividing the space into left and right (see also ref. 23 for visual representations of how this works for spatial navigation in rodents). How can we test this? First, we extracted activity patterns for the 4 quadrants separately for the two contexts. Then we looked, using a whole-brain searchlight, for brain

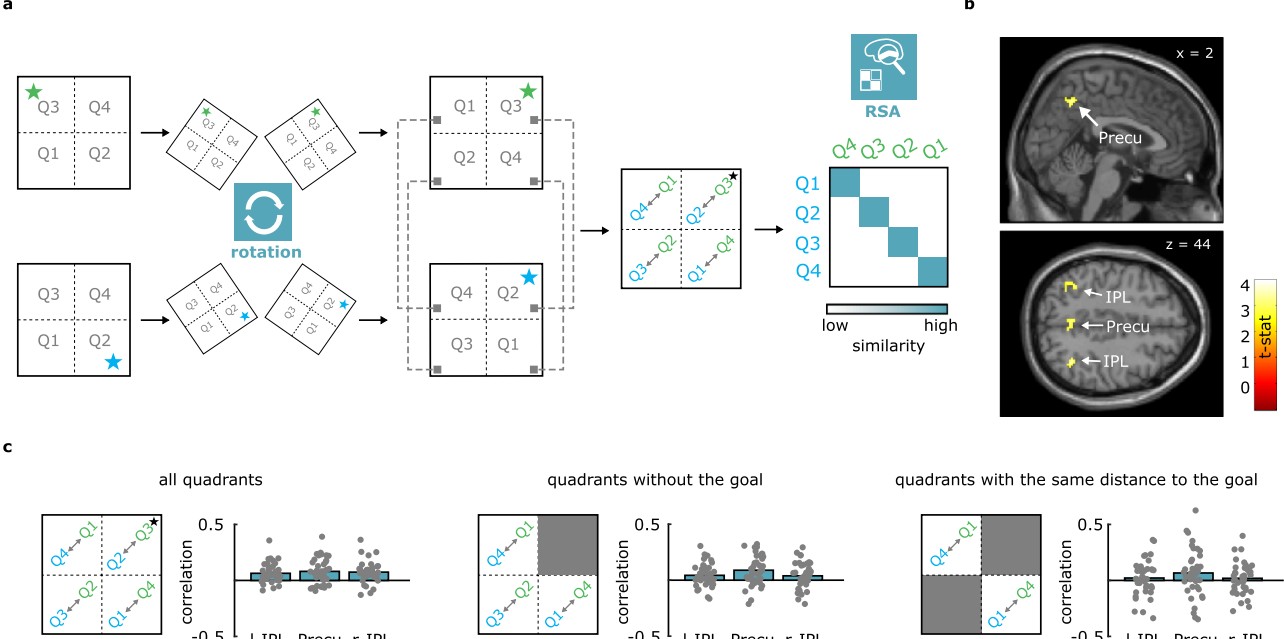

**Fig. 3 | Evidence for a goal-directed mental rotation of the feature maps in the parietal cortex. a** Logic of the rotation analysis, where quadrant-by-quadrant similarity was measured after assuming a rotation of one of the two spaces (see Methods). **b** Whole-brain results of the correlation analysis, one-sided *t*-test as implemented in SPM12, thresholded at $p < 0.001$ at the voxel level, corrected for multiple comparisons at cluster level with $p < .05$. **c** Control analyses (two-sided *t*-test) confirm the effect is not driven by goal quadrants alone (middle panel and Supplementary Fig. 16), and that this is not, at least in the precuneus, a mere effect of distance from the goal (right panel) ($n = 40$). Please note that for this last control analysis, no effect was detected at an unbiased whole-brain level, not even at a threshold of $p < 0.05$, probably because of the low number of remaining trials, as the same happened when we considered other pairs of quadrants only.

regions that had an increased similarity between the quadrants in one context (e.g., blue molecule) with the corresponding quadrants in the other one (e.g. green molecule) after the feature space was rotated by 180°. In other words, after we aligned the goal locations in both contexts according to the model just described (e.g., the activity evoked by trials occurring in quadrant 1 (Q1) for the blue molecule would be more similar to that evoked by trials in Q4 for the green molecule, see Methods and Fig. 3a)."

The results revealed significant clusters in the precuneus ($MNI_{x,y,z} = -2 -56 46$, $t = 4.5$) and in the bilateral IPL (left: $MNI_{x,y,z} = -32 -62 40$ $t = 4.93$; right: $MNI_{x,y,z} = 36 -56 42$, $t = 4.89$, see Fig. 3b), that comprise the brain regions previously reported to be involved in mental rotation of visually displayed objects[47,48] as well as egocentric processing[32,33]. This effect could not be explained as a simpler cross-context correspondence of goal-quadrants (vs. non-goal quadrants) because it was still present when we removed them from the analysis (Fig. 3c middle panel and Supplementary Fig. 14), nor as cross-context similarity of distance from the goal, because in the precuneus it remained significant (although to a weaker extent) also when we only considered Q1 and Q4, that were at the same distance from the goal (Fig. 3c right panel). This finding suggests that the parietal representation of conceptual spaces is transformed and re-oriented to maintain the focus on goals and their relation with our current state, independently of the global map.

## Evidence of cross-context grid-like representations in entorhinal cortex

During spatial navigation, egocentric representations are complemented by a parallel reference frame, reflected in allocentric map-like models of the navigated environment[5]. Such "cognitive maps" in mammals depend mostly on the activity of the hippocampal–entorhinal circuit. A key neural signature of an allocentric map is entorhinal grid cells[8]. In fMRI, a proxy measure of grid cell population activity can be observed: the fMRI signal is systematically modulated as a function of movement direction (hexadirectional activity[9]), where trajectories that are at a multiple of 60° apart from each other ("aligned") are represented more similarly than those that are not ("misaligned")[45]. Thus, to investigate the effects of conceptual goals on allocentric cognitive maps, we first looked for signatures of the grid-like code. We implemented a variation of the grid-RSA approach previously employed by our groups[17,49] and others[14]: for each trajectory in one context (e.g., green molecule), we assumed higher pattern similarity in the entorhinal cortex for those trajectories in the other context (e.g., blue molecule) that were aligned to multiple of 60°, and lower pattern similarity for those that were misaligned (see Fig. 4b for the predicted model. Note that the egocentric mental rotation reported above would not affect this analysis, since a rotation of 180° is a multiple of 60°). We found that (i) the similarity was higher for aligned compared to misaligned trajectories in the left posterior-medial entorhinal cortex (left pmEC, mean Pearson's $r = 0.03$, std $= 0.10$, $p = 0.012$; right pmEC: mean Pearson's $r = 0.02$, std $= 0.10$, $p = 0.092$, Fig. 4c), (ii) that this effect was not statistically significant in the anterolateral entorhinal cortices (all $p > 0.91$), and (iii) that control periodicities did not show statistically significant effect (all $p > 0.09$, Fig. 4d). We found the 6-fold modulation in the left entorhinal cortex also using a whole-brain searchlight procedure ($MNI_{x,y,z} = -12 -14 -28$, $t = 3.36$, $p < 0.005$ at voxel level, uncorr., Fig. 4e). Thus, we observed a signature of allocentric maps, namely grid-like representations in the entorhinal cortex, during our conceptual task, concurrently with the egocentric-like signatures in the parietal lobe, as predicted in ref. 6.

## Evidence for goal-induced alteration of grid-like representations in the entorhinal cortex, medial prefrontal cortex, and superior parietal lobule

Interestingly, recent evidence suggests that goal locations alter rodents' grid-like maps during spatial exploration[38,39]. We reasoned that this mechanism if paralleled during the conceptual search, might

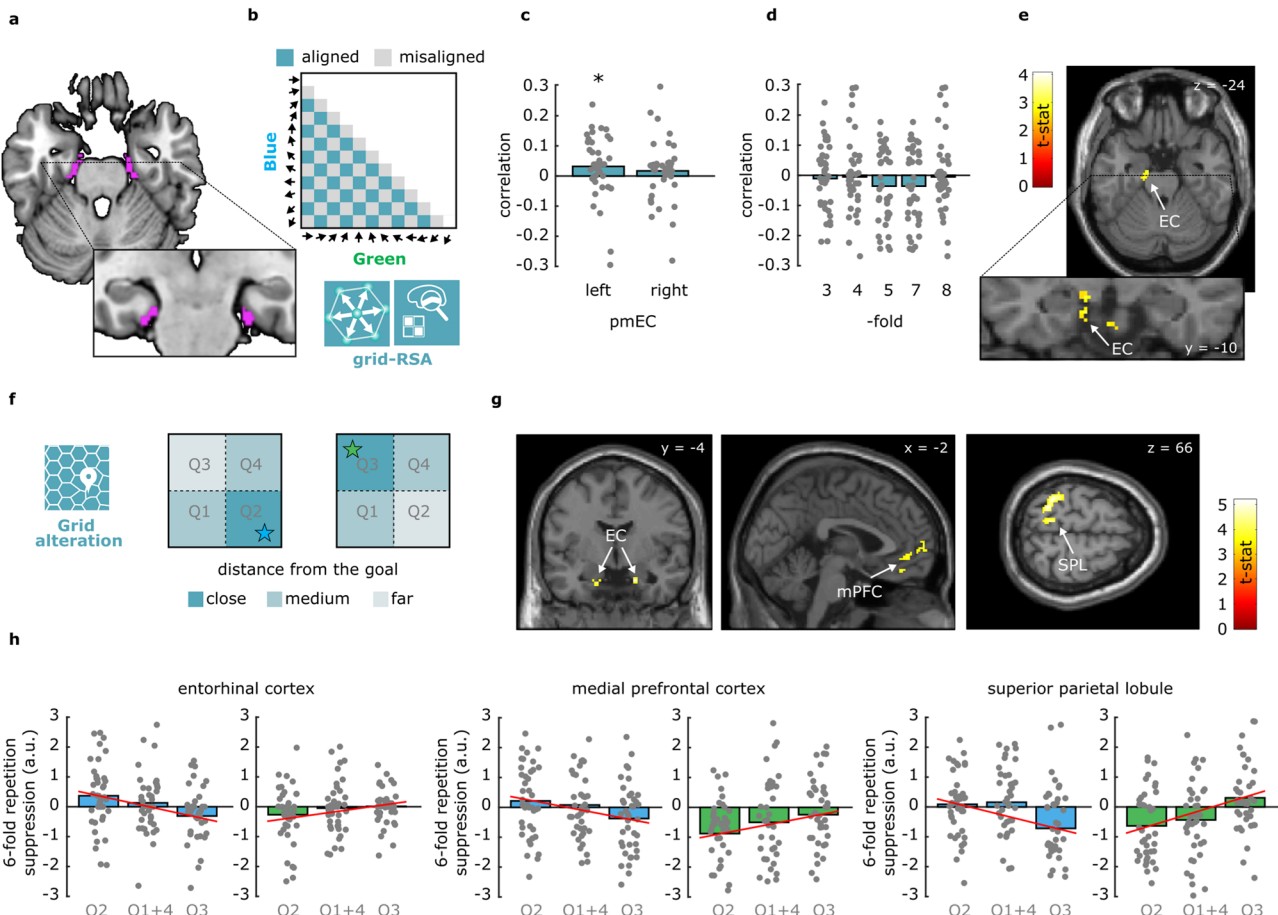

**Fig. 4 | Evidence of grid-like coding in left pmEC and of its alteration in right pmEC, mPFC, and SPL. a** ROIs of posteromedial entorhinal cortex (pmEC) taken from ref. [62]. **b** Logic of the cross-context grid-RSA, where patterns of activity evoked from morphing trajectories in one context are correlated with those in the other one assuming a 60° rotational similarity (e.g., moving at 30° in the blue context would be more similar to 90° in the green context than 0°). The model matrix shows aligned and misaligned pairs, for which we predicted higher and lower similarity, respectively. The diagonal is excluded to avoid biased effects of homologous directions across the two contexts. **c** A significant ($p < 0.02$) grid-like effect is observed only in the left pmEC ($n = 40$, two-sided $t$-test). **d** None of the control symmetries showed a significant effect in the left pmEC ($n = 40$). **e** Whole-brain results of the same effect, here shown thresholded at $p < 0.01$ uncorr. for visualization purposes. **f** Logic of the goal-related analysis, where we expected difference in the grid-like code as a function of the quadrant, that is, as a function of goal distance. **g** Whole-brain results of one-sided $t$-test as implemented in SPM12, thresholded at $p < 0.001$ at the voxel level, corrected for multiple comparisons at cluster level with $p < 0.05$ (EC small volume corrected with bilateral pmEC mask shown in panel (**a**)). **h** visualization of the linear effect across quadrants and contexts, indicating that the alteration followed the goal position across the two spaces, blue and green. Q1 and Q4 are averaged, as they are equidistant from the goal. Please notice that averaging before running individual subject-level glm or after that did not affect the results. $n = 40$. a.u. arbitrary units.

support the readout and access of conceptual goals, as it would potentially differentiate behaviourally relevant regions of conceptual spaces from irrelevant ones. Therefore, we asked whether the grid-like code is modulated as a function of distance to conceptual goals. To examine this question, we used a different approach, partially inspired by Doeller and colleagues[9], that allowed us to conduct a more fine-grained quadrant-specific grid analysis as a function of goal proximity. More specifically, we followed a repetition suppression approach where each trajectory was assumed to evoke a signal that scales as a function of the angular distance in 60° rotational space compared to the previous one. For instance, a trajectory at 90° would elicit a weak (suppressed) signal if presented after one at 30° because they would be 60° apart, thus perfectly aligned in the 6-fold rotational space, while the same trajectory would elicit a stronger signal if preceded by one at 0° because they would be 30° apart in the 6-fold rotational space. Following our hypothesis, we modeled the grid-like modulation separately for the four quadrants, where trajectories were evenly sampled and distributed (see Methods). Among these four quadrants, one contained the goal, and was considered as at a "close distance" from it (e.g., Q2 in blue context), two were at the same "medium

distance" from the goal (e.g., Q1 and Q4 in blue context, and were thus averaged) and one was at a "far distance" from the goal (e.g., Q3 in blue context). If allocentric grid-like maps are involved in representing goal locations during conceptual navigation, we expect the grid-like signal to be modulated as a function of the quadrant, that is, as a function of the distance from the goal. More specifically, given the above-mentioned empirical evidence in rodents of increased grid-cell activity around goal locations (both in terms of firing rate and number of grid fields), we expected these putative regions to show a weaker grid-like signal for trajectories happening in quadrants far from the goal (Q3 in the blue context, Q2 in the green context), a medium grid-like signal for those happening in quadrants at medium distance from the goal (Q1 and Q4, identical for the two contexts), and a stronger grid-like signal for the trajectories in goal quadrants (Q2 in the blue context, Q3 in the green context) (see Methods and Fig. 4f). A whole-brain analysis revealed this modulation in regions where grid-like signals have been previously observed for navigation in both virtual reality and conceptual spaces[12,15]: the right entorhinal cortex (MNI$_{x,y,z}$ = 16 −6 −26, $t = 4.23$, $p < 0.001$, small volume corrected with bilateral EC mask), the medial prefrontal cortex (mPFC; MNI$_{x,y,z}$ = 2 48 −2, $t = 4.46$, $p < 0.001$

corrected for multiple comparisons at cluster level with $p < 0.05$), and the superior parietal lobule (SPL; MNI$_{x,y,z}$ = −30 −46 66, $t$ = 5.18, $p < 0.001$ corrected for multiple comparisons at cluster level with $p < 0.05$; Fig. 4g): here, the grid-like signal changed as a function of goal proximity, being relatively weaker for trajectories that were far apart from goal locations and stronger close to the goal, see Fig. 4h. An additional, weaker cluster was observed in the left angular gyrus, see Supplementary Table 3. Taken together, these results indicate that a typical signature of the cognitive map, the grid-like code, is modulated by the presence of conceptual goals.

## Discussion

Learning how the human brain supports access to stored conceptual information is important for understanding our cognitive abilities. A fascinating hypothesis put forward in recent years is that a phylogenetic continuity exists between the brain's spatial navigation circuits and the human conceptual system, with concepts memorized as points of internal cognitive or conceptual maps in the medial temporal lobe[3,18]. The discovery that humans can recruit the hippocampal formation to represent how concepts in memory relate to each other supports this proposal[12–17], but the human spatial navigation system comprises an extended network of brain regions beyond the medial temporal lobe, that represents space via complementary allocentric and egocentric frames of reference[21–23]. Recently, we have hypothesized that this fully integrated network might support conceptual "navigation" across these different reference frames, for instance guiding us during mental search[6]. In line with this hypothesis, in the current study, we reported three main findings, showing the parallel recruitment of complementary reference frames to track conceptual goals during mental search. First, we showed evidence of egocentric-like codes in the parietal cortex to represent goal position in conceptual spaces relative to our current state or experience. Intriguingly, we also observe preliminary evidence for egocentric representations reflected in the pattern of eye movements: participants were moving their eyes to the left or to the right consistently with the position of the goals in conceptual space, thus in line with the idea that they were mentally searching for concepts using a first-person, self-centered, perspective. Second, we showed that the representation of the two conceptual spaces in the parietal cortex was rotated to match goal positions across contexts. Finally, we showed that the brain supports mental search in conceptual spaces by concurrently altering the structure of the allocentric grid-like code in medial temporal, prefrontal, and superior parietal cortices as a function of goal proximity. Taken together, these representational schemes might provide critical information about "where" goals are stored in conceptual spaces, employing both egocentric and allocentric reference frames in parallel.

According to an influential model of spatial memory and imagery[21–23], the parietal cortex and the medial temporal lobe are the two major hubs of our spatial navigation system, providing a representation of the external navigable space across different reference frames. In this perspective, the parietal cortex holds egocentric codes that capture the locations of objects and landmarks in self-centered coordinates (e.g., whether an object is to our left or to our right), as demonstrated by several empirical findings[28–36], see Introduction. In our study, we reported remarkably similar observations in the parieto-occipital sulcus (POS), the precuneus, and the PPC extending to the RSC. These regions appeared to represent goal locations in conceptual spaces according to a reference frame that was independent of the position of the goal across contexts, the allocentric angle of the trajectory, and the current allocentric position in space. This representational scheme can be interpreted as an egocentric vector relative to the goal: Akin to the physical world, where we can tell whether an object is to our left or right while we move and change position in the environment, here, by considering how the bond-length ratio of the

molecule stimuli changed, we could define to what degree a goal concept was on one side or the other in the underlying feature space, irrespective to its external geometry. This indicates the recruitment of egocentric-like schemes in the parietal cortex for representing conceptual goals. The role of the parietal cortex beyond spatial cognition has been previously reported in studies on conceptual/semantic access and autobiographical memory (see refs. [50–53]). They have suggested the precuneus and the inferior parietal lobules as key regions of these networks. However, a direct link between the representational codes typically evoked in this region during spatial navigation and more abstract conceptual processing has been formulated only recently, with fMRI studies revealing similar parietal activity when participants have to evaluate the spatial proximity of locations, the "temporal proximity" of events and the "emotional proximity" or people, from an egocentric point of view (namely, how far/close from one's own self; see refs. [54–57]. The present results go beyond these previous reports by revealing egocentric vector coding as a function of goal location during conceptual navigation. Supporting this, the additional, preliminary, evidence coming from the analysis of eye movements using DeepMReye suggested that participants were employing a first-person perspective of the conceptual spaces while mentally searching through them. Although the DeepMReye results tend to support the egocentric spatial interpretation of mental search in our task, they also raise a puzzle in that they suggest that the large majority of participants must have adopted the same axis orientation for this space; the axis orientation (e.g., upper bond-length increases to the right in Fig. 1b) is in fact arbitrary, so the presence of a consistent effect among participants is perhaps surprising, and might reveal the existence of internally biased representations for conceptual spaces similar to those observed in the physical environment (e.g., the left-to-right orientation in enumeration or reading in most of the Western cultures). In sum, our findings support the idea that humans represent goals in conceptual spaces using egocentric reference frames, relative to our specific self-centered perspective and experience, beyond the spatial domain, as theoretically predicted in our recent account of the cooperation between parietal and medial temporal regions in conceptual knowledge representation across reference frames[6].

This account also stresses that, besides egocentric representations in the parietal cortex, physical environments are also represented in our brain using a complementary coding scheme tuned to an allocentric perspective, namely viewpoint-independent representations of the general layout of the environment[21,22]. A typical signature of allocentric representations in the brain is the grid-like code, likely originating from the population activity of grid cells, and typically observed in the medial entorhinal and prefrontal cortices in humans engaged in both virtual and conceptual navigation[9]. Interestingly, two recent studies reported that entorhinal grid cells in rats change their firing pattern in the proximity of goals, where an increased number of firing fields as well as a reduced distance between them was observed[38,39]. Whether this also applies to human grid-like representations during spatial navigation is still unclear, especially given the difficulty of drawing direct comparisons between individual neural activity patterns in rodents and indirect neuroimaging measures in humans (but see[20]). Nevertheless, we reasoned that such a mechanism would be, in principle, useful for the readout of conceptual goals from cognitive maps, as it would provide an allocentric signature of goal positions in conceptual spaces that is complementary to parietal egocentric-like codes. By implementing a grid-like analysis as a function of distance from the goal, we indeed showed that the grid-like signal in the right entorhinal cortex, in the medial prefrontal cortex, and in the superior parietal lobule was weaker for regions of the conceptual environments that were not behaviourally relevant, while it was relatively stronger for regions containing a goal. This indicates that the grid-like code might not provide a mere metric of space, physical or conceptual, that reflects the general layout of the navigable environment, but it might

also be recruited for the representation of task-relevant conceptual regions[19], such as specific goal concepts (see ref. 58 for recent fMRI evidence of goal-induced alteration of hippocampal cognitive maps) .

Whether and how this is related to goal-directed, egocentric-like responses in the parietal cortex remains unclear. According to models of spatial navigation and memory[22,23,56], allocentric representations in the hippocampal-entorhinal system interact with the egocentric ones in the parietal lobe via the medial parietal cortex and, in particular, the retrosplenial complex (RSC), with information flowing in one direction or the other on the basis of task demands. For example, when we mentally construct the general allocentric layout of a visited environment from different perspectives, the information is supposed to be directed from parietal to medial-temporal cortices; inversely, when we recover from memory a specific image or scene of an episode, the information would flow from the medial-temporal lobe to the parietal cortex. In both cases, the RSC is thought to provide a transformation of the reference frame (from egocentric to allocentric, or vice versa[22,23]). Although we did observe activation in the PPC/RSC in our decoding analysis, the exact role of this region and whether it was involved in transforming one reference frame into the other during our conceptual navigation task could not be determined. It is possible that the alteration of the grid-like map is beneficial for orienting the focus of our "parietal window" (corresponding to the observer in spatial navigation tasks, according to models of spatial memory[22]) towards the goal, thus resembling the hypothesized top-down effect proposed for reconstructing specific egocentric scenes from allocentric maps[22]. This would be in line with recent evidence that perturbing the grid-like code in rodent entorhinal cortex impairs egocentric-like representations in the RSC[57], but whether this also applies to the access to conceptual knowledge, or the underlying computational principles that allow the brain to operate this transformation, is unknown. Future studies could address the interplay between the parietal cortex and the medial temporal lobe during conceptual search by investigating, for instance, the connectivity profile between the two regions as a function of the task at hand.

Finally, in our study we also reported an intriguing additional observation, namely the rotation of the representation of the two contexts in both the medial and the lateral parietal cortices: in these regions, the representations of the two contexts were highly similar when they were rotated in such a way that goal positions (or quadrants, in our case) were perfectly aligned. This effect is reminiscent of the well-known phenomenon of mental rotation, which is dependent on the parietal cortex, as demonstrated by both neuroimaging and non-invasive brain stimulation[46–48], and has been proposed as a candidate mechanism for viewpoint-dependent orientation[45]. Thus, one interpretation of this finding is that conceptual spaces might be rotated, or structurally aligned, in the parietal cortex to maintain the center of the "parietal window" on the task-relevant conceptual goals and that this might be a crucial stage in the process of orienting and transforming egocentric representations into allocentric ones[21–23]. At the present stage, this interpretation remains speculative, and further investigations are needed to provide empirical support to this hypothesis. Yet it also generates further intriguing speculations on the role of mental rotation abilities as a precursor of more abstract and sophisticated forms of "conceptual rotation", for instance in social settings, where we need to change "our perspective" to better understand others' positions and mental states. Additionally, it should be acknowledged that the process of mental realignment between the two contexts might have been facilitated and made more spontaneous by the symmetry of the visual stimuli used, and new studies should address whether and how competitive visual factors (e.g., asymmetrical bonds in our molecules) affect the results.

To conclude, we reported evidence that conceptual goals are encoded via egocentric-like representations in the parietal cortex and that they alter the allocentric grid-like map in the medial temporal and prefrontal cortices. Whether these results extend to non-human species as well (as for instance happens for allocentric map-like coding[20]) cannot be concluded with the current experiment, but our findings can contribute to our understanding of how humans organize and search for conceptual information in memory, and further support the proposal that the brain's navigation system can be repurposed to represent knowledge across different reference frames[6].

## Methods

### Participants

Participants were 40 adults (19 self-identifying as females, 21 as male) with mean age = 27.2 years (std = 4.8). They all gave informed consent and were reimbursed for their time with 9 euros/hour for behavioral tasks and 10 euros/hour for the fMRI scanning session. All had normal or corrected-to-normal vision, no history of neurological diseases, and were right-handed. Differences in sex or gender were not considered in our analyses because they were out of the scope of the experiment. The sample size was determined without a priori power analysis, due to the small number of experiments conducted in conceptual navigation and no existing research testing specifically the egocentric code in concept spaces. Moreover, the study involved several different types of analysis (fMRI adaptation, multivariate decoding, grid-like analyses), in addition to behavioral analyses, which further complicated the determination of an effect size of reference in previous literature. In an attempt to decide the relatively appropriate sample size, we took guidance from studies in this area which have drawn informative results from smaller pools of participants. For example, in studying goal direction signals in a spatial task, Chadwick and colleagues (2015) included 16 adult participants. In a study that was very influential in the development of our own (but that focused on allocentric coding schemes), 21 participants performed similar tasks[12]. Based on these considerations, we refrained from forming precisely quantified expectations and aimed to test 40 participants, which was roughly twice the number tested in the majority of experiments in this area that consistently showed similar results for allocentric and egocentric navigation. The study did not include between-group analyses, thus no blind assignment was performed. The study was approved by the local Ethics Committee of Leipzig University, Germany (protocol number 159/21-ek).

### General experimental design and cover story

The experiment consisted of 3 days. At the beginning of the first day, after signing the informed consent, participants were told the cover story of the experiment: they were asked to play the role of scientists in a future society where mankind lives in peace and prosperity thanks to the development of two molecules, called Megamind and Megabody, able to enhance cognitive and physical abilities, respectively. Molecules were fictitious and just represented visual stimuli on the computer screen during the entire experiment: participants did not interact with real biological molecules or chemicals of any sort. They were told that, for the next 3 days, their task would be to learn how to produce these molecules by manipulating specific reactions through a computer program in order to respond to the demand for supply in society.

During the first 2 days, participants performed computer-based behavioral training in a laboratory setting using a monitor PC in order to familiarize themselves with the feature spaces and their conceptual goals. During these two days, they performed three different tasks, named "Collect task with the hint", "Collect task without hint", and "Recall task" (see below). Day 1 lasted roughly 1 h and 30 min, and day 2 lasted ~ 1 h, as participants were already familiar with the tasks and procedures and performed faster and better (see Results in the main text). On the third and last day of the experiment, participants performed the Recall task while their brain activity was monitored using fMRI. This last procedure lasted about 2 h.

## Stimuli

We created two visual stimuli that we called "molecules" on Microsoft PowerPoint (Microsoft Corp.). These so-called molecules are just hypothetical toy examples detached from real-world chemistry. Our stimuli had a central body and two extreme parts that were connected to it by two bonds, an upper and a lower one. Throughout the experiment, different molecules could vary in their "bond-length ratio", namely the relative length of one bond to the other. Crucially, by considering the length of the two bonds as axes of a two-dimensional space, we could conceive each configuration (that is, a molecule with a specific bond-length ratio) as a point in the two-dimensional space. Molecules were created in two colors: blue and green, which indicated the two different contexts. Participants learned that the molecule named "Megamind" was a specific configuration of the blue molecule, with a longer upper bond and a shorter lower one, and that the molecule named "Megabody" was a specific configuration of the green molecule, with a shorter upper bond and a longer lower bond (Fig. 1a). Blue molecules with bond-length ratio different from the one of Megamind and green molecules with bond-length ratio different from the one of Megabody were considered "wrong configurations" and did not have any behavioral relevance in the experiment (this also applies to the blue molecules with the same bond-length ratio of Megabody and the green molecules with the same bond-length ratio of Megamind).

## Collect task with hint (behavioral training)

This task was administered at the beginning of both training days. Participants were prompted with a screen showing several elements. In the central part, they were presented with a colored molecule in a given random configuration. On the right, they had a symbol which we referred to as the "hint": this symbol was gray when the molecule shown on the screen was in the wrong configuration, and it turned blue or green when the molecule was correctly morphed in the goal configuration. On the left, they were presented with a controller composed of two vertically graded bars. Participants were told that the physical attributes of the molecule (the bond-length ratio) could be changed by manipulating the amount of oxygen and hydrogen (aka the reagents) in the surrounding of the molecule: increasing the oxygen or the hydrogen would make the upper or lower bond, respectively, longer; decreasing them would make them shorter. The reference to the reagents was part of the cover story and they were not really manipulated in the experimental room. Participants were instructed to press the "Z" or the "Y" (German keyboard) key on the keyboard to adjust the ratio of change they wanted to apply to the molecule: by doing so, the computer displayed a different relative amount of reagents on the controller. Z and Y controlled the relative ratio in opposite directions, which could be interpreted as changing the direction vector clockwise or anti-clockwise. In spatial terms, this was conceivable as setting a facing direction of movement. Holding the "Z" key, for instance, would continuously reset the ratio between hydrogen and oxygen, and would correspond to continuously rotating the facing direction in 360°. In other words, the hydrogen and oxygen corresponded to the sine and cosine projections of the "facing direction" vector once considered on a trigonometric circle.

Then, when the choice was made and the participant felt confident, they could apply the change to the configuration that was presented on the screen at that specific moment, by pressing "C" or "V". Holding these keys would allow the morphing/movement to continue along the same trajectory ("C") or in the opposite direction ("V"). Two horizontal bars, placed above and below the molecule at fixed distances from the central body, indicated the limits for the configurations: the two bonds could not be elongated beyond these points. Similarly, they could not be set shorter than the position of the central body. In spatial terms, these represented the boundaries of the two-dimensional spaces that were implicitly navigated. Participants were instructed to try different combinations of oxygen/

hydrogen ratios to find the correct morphing trajectory that could lead, when applied, to the goal configuration. When the morphing molecule corresponded to the correct configuration, the hint symbol turned blue or green and participants could collect the molecule by pressing the spacebar on the keyboard. In this task, this key was disabled for wrong configurations, meaning that participants could not collect the wrong molecules. Subjects had 30 minutes maximum to collect 20 Megamind and 20 Megabody, with blue and green molecules presented pseudo-randomly trial after trial. A small counter placed below the hint symbol informed them how many molecules out of 40 they had collected so far. Example trials are available in Supplementary Movie 1.

## Collect tasks without hint (behavioral training)

This represented the second task participants performed on both training days. The visual appearances as well as the general demand were identical to the previous one, except for a few pivotal differences. First, the hint symbol was removed: participants had to recall the correct bond-length ratio of the molecules. Second, they could collect wrong molecules, which counted as errors. Third, they performed this task separately for the blue and green molecules. Fourth, and last, they were instructed to collect as many goal molecules of a given color as possible in 3 minutes. The order of the two contexts was counterbalanced across the two days.

## Recall task (behavioral training and fMRI)

This was the last task on both training days. Participants were told that they had to evaluate some of the reactions they had tried in the previous tasks in order to classify them as efficient or not to produce the goal molecules. Reactions were instead new, and created to satisfy the specific requirements of our design (e.g., balance between directions, quadrants, and so on). They performed two runs, 1 with blue molecules, and 1 with green ones. For each trial, a molecule in a random configuration was shown at the center of the screen, without any other detail except for the upper and lower horizontal boundary lines. The molecule remained on the screen for 0.5 s, then it started morphing automatically for about 1 s, before stopping again and remaining on the screen with the updated configuration. At this point, the "imagination period" started: participants were instructed to imagine the morphing to continue in the same way, following the same changes in the bond-length ratio, and to decide whether such a morphing would ever result in the molecule matching the correct goal configuration of Megamind (for blue molecules) or Megabody (for green molecules). In spatial terms, this was conceivable as imagining the movement to continue along the same trajectory and deciding whether the goal location would be reached or missed. After 4 s of imagination, participants were prompted with a question asking whether they would ever obtain the correct configuration if the molecule continued to morph in the same way, and they could select one of two answers (Yes or No, with their left and right assignment on the screen balanced across trials). After the answer was given, the following trial started. This "target question" happened on a pseudo-randomly selected 50% of trials. In the remaining half, we introduced a "filler question" where, at the moment when the question was shown, we displayed a random molecule on the screen that could be "on the trajectory" or not, and we asked whether they would ever obtain *that particular* molecule if the morphing continued in the very same way (Fig. 2c). This question was introduced to induce participants to create a global map of the navigable spaces while holding in mind the conceptual goals so that we could test concurrent representation of allocentric cognitive maps in the brain: given that (i) participants did not know in advance the kind of question they were going to be asked, and (ii) that we analyzed the imagination period preceding the question (see below), we ensured the optimization of chances to detect all of our effects of interest. Participants performed the recall task also during the fMRI scanning

session. The procedure remained identical, except for the fact that there were 8 runs (4 with blue molecules, 4 with green molecules, intermixed).

There were 48 trials per run, divided in the following way. Irrespective of the question asked, morphing trajectories were directed to the goal (Megamind or Megabody, catch trials) 25% of the time (12/48 trials). The remaining 75% of trials (36/48) were equally divided into 6 egocentric-like conditions, missing the goal with angular distances to the correct trajectory of −135°, −90°, −45°, +45°, +90°, +135°. The sign (+ or −, arbitrarily defined) of these angular trajectories determined whether the goal was missed on the "left" or on the "right". As the target question happened on 50% of the trials, participants were expected to answer Yes to this question on 6 trials, and No on 18, for a total of 24 trials (24/48 = 50%). In the remaining trials, when the filler question was asked, we generated molecules for the question period that could lie on the current trajectory (34% of the time), or not (33% of the time it was out of trajectory by -60°, the other 33% of the time it was out of trajectory by +60°). Given the complex and noncorresponding percentage of trials and expected responses, performance for this task was expressed as the probability of Hits (probability of saying Yes when Yes was the correct answer) and False alarms (probability of saying Yes when No was the correct answer). Trials were equally divided into the 4 quadrants composing the feature space, by considering the midlines of both axes as boundaries. This allowed us to obtain a very small correlation across trials between egocentric angles to the goal and distance from the start position to the goal (Pearson's $r = 0.03$, p = 0.5). Control analyses described in the Supplementary Materials confirmed that minor confounding factors did not explain our results. No morphing period implied a crossing of between-quadrant boundaries.

### fMRI data acquisition

fMRI data were acquired at the Max Planck Institute for Human Cognitive and Brain Sciences (Leipzig, Germany) using a 3-Tesla Siemens Skyra scanner equipped with a 32-channel head coil. High-resolution T1-weighted images for anatomical localization were acquired at the end of the scan session using a three-dimensional pulse sequence with TR = 3.15 ms; TE = 1.37; flip angle = 8; voxel size = 1.6 × 1.6 × 1.6 mm). T2*-weighted images sensitive to blood oxygenation level-dependent (BOLD) contrasts were acquired using a pulse sequence with TR = 1500 ms; TE = 22 ms; flip angle = 80°; voxel size = 2.5 × 2.5 × 2.5 mm; multiband acceleration factor of 3. Participants viewed the stimuli on the screen through a mirror attached to the head coil, and behavioral responses were collected using a button box.

### Preprocessing

Images were preprocessed using a standard pipeline with SPM12, which included slice-time correction, realignment of functional images to the middle scan of each run, coregistration of functional and anatomical images, segmentation, normalization to the Montreal Neurological Institute (MNI) space, and smoothing of 5 mm isotropic.

### Egocentric-like adaptation analysis

The first analytical approach that we implemented was a univariate adaptation analysis, building on the observation that the fMRI BOLD signal shows suppression (or adapts) when stimuli are repeated, potentially because of the adaptation of the underlying neuronal populations[59,60]. We reasoned that if a brain region is representing specific egocentric conditions differently, then it should show a suppression pattern specific to each individual condition. More specifically, we defined a first-level generalized linear model (GLM) for each participant by modeling, in one regressor, the onset of the imagination period for each trial, and modeling trial duration for the entire imagination period. Following previous studies successfully modeling

directional representations in the brain[9,17], we applied a parametric modulator to this regressor where we included, for each trial, the (log) time passed since the last presentation of the same egocentric-like condition (e.g., since the last time that a trajectory with the goal at −45° was presented): in other words, we assumed that a brain region that represents separately the egocentric conditions would show release from adaptation that is modulated by the temporal recency of the specific condition (Fig. 3b). Trials with the first presentation of a given egocentric condition were excluded. Additional regressors of no interest (included also in all the subsequent analyses) modeled the time of response (1 regressor) as well as movement parameters (6 regressors). Group-level (2nd level) analyses were conducted using SPM12 by running voxel-by-voxel $t$-tests of these contrasts across subjects. Corresponding analyses were performed when controlling for additional factors, such as removing catch trials (all the egocentric-like conditions directed to the goal), separating runs for blue and green molecules, or modeling left vs right (see main text for the full list).

### Cross-context decoding of egocentric-like conditions

The second analytical approach that we implemented was a multivariate decoding analysis (also known as Multivoxel Pattern Analysis or MPVa[40]): here, instead of analyzing the univariate BOLD change at the single voxel level, the distributed activity pattern in a specified region is considered, and each voxel provides its own contribution for training (and then testing) a multivariate classifier incorrectly decoding the presented condition. Thus, we ran a second GLM, modeling each egocentric-like condition separately (trajectories with a direction −135°, −90°, −45°, 0°, +45°, +90°, +135° to the goal). We then used the MATLAB (Mathworks Inc.) toolbox COSMoMVPa[61] to run a searchlight cross-context decoding approach. We defined spheres of radius of 3 voxels (consistent with previous studies (e.g.,[16]) centered on each voxel of the brain and, within these regions, we extracted the activity pattern for each egocentric-like condition, separately for the two molecules. We used a Nearest-Neighbor (NN) classifier as implemented in the COSMoMVPa toolbox to distinguish the 7 egocentric-like conditions in one context (e.g., blue or green) after training it on the activity patterns evoked in the other context (e.g., green or blue), using a cross-validation scheme. From the accuracy level, we subtracted the chance level (1/7 = 14.29%) and we stored the resulting performance of the classifier at the center voxel of each sphere, creating one brain map per subject. Group-level analysis was performed with SPM12. Corresponding analyses were performed by excluding catch trials (egocentric-like conditions of 0°) and considering a chance level of 1/6. Corresponding results were obtained using a linear discriminant analysis (LDA) classifier as implemented in the COSMoMVPa toolbox.

### Rotation of the feature spaces

We ran a GLM with 4 regressors of interest, each one modeling trials that were happening in one of the 4 quadrants of the conceptual space, separately for the two molecules (Q1 blue, Q1 green, Q2 blue, Q2 green, and so on). We used COSMoMVPa to run a whole brain searchlight (spheres with radius 3 voxels) using a cross-context correlational approach. We constructed a similarity matrix where the activity patterns separately evoked by the four quadrants in one context (Q1 blue, Q2 blue, Q3 blue, Q4 blue) were correlated (Pearson's $r$) with those in the other one in reversed order, as if the space was rotated 180° (Q4 green, Q3 green, Q2 green, Q1 green). We looked for brain regions where correlation values on the diagonal (representing the correlations Q1 blue → Q4 green; Q2 blue → Q3 green, Q3 blue → Q2 green, Q4 blue → Q1 green) were higher than off-diagonal, storing the Fisher's-to-z transformed on- vs. off-diagonal difference in the center voxel of each sphere and creating one map per subject, that were later analyzed at the group-level using SPM12.

## Cross-context grid-RSA

To investigate the presence of a grid-like signal during goal-directed conceptual navigation, we implemented a variation of the grid-RSA[14,16,49]. We first resampled all the allocentric movement trajectories in bins of 30° with the direction 0° arbitrarily aligned to the horizontal axis. We run a GLM modeling the resulting 12 conditions (0°, 30°, 60°, 90°, 120°, 150°, 180°, 210°, 240°, 270°, 300°, 330°) separately for the two contexts, and we then moved to multivariate analyses within the COSMoMVPa environment. We constructed a cross-molecule similarity matrix, where the activity patterns of each of the 12 allocentric directions from one context were correlated (Pearson's r) with those from the other context. The resulting matrix was made symmetrical by averaging the lower and upper triangles, and the diagonal was removed to ensure that any potential grid-like effect was not driven by the higher similarity between homologous directions across the two environments. We constructed a predicted model which assumed, for each movement direction in one context, higher similarity with those movement directions in the other context that was at multiples of 60° from it (Fig. 4b). For example, moving at 30° in the blue context would elicit higher similarity with allocentric movement directions of 90°, 150°, 210°, 270°, and 330° in the green context, that were referred to as "aligned", compared to the remaining ones ("misaligned"). We thus correlated (Pearson's r) the lower triangle of the neural similarity matrix with the lower triangle of the model and stored the Fisher's-transformed correlation values for each subject. We applied this analysis at first in a Region of Interest in the posteromedial entorhinal cortex, and in control regions in the anterolateral entorhinal cortex, using masks from ref. 62. We then applied the same analysis in a whole-brain searchlight, using spheres with a radius of 3 voxels. Control models were developed in the same way but assuming rotational symmetries of 120° (3-fold), 90° (4-fold), 72° (5-fold), 51.4° (7-fold), and 45° (8-fold).

## Alteration of the grid-like code

To investigate the possibility of an alteration of the grid-like code, we proceeded using a repetition suppression approach, that allowed us to analyze the grid-like code as a function of goal proximity. Our trials were equally divided into 4 quadrants: one of them contained the goal, two were at a medium distance from it, and the last one was far from the goal. Importantly, the goal quadrant differed across the two contexts. We ran a GLM where we modeled separately the trials in the four quadrants with 4 regressors and applied a parametric modulator to each of them where we modeled the activity evoked by each trial as a function of the angular distance in 60° rotational space from the previously presented trial. This analysis was a modified version of the grid-adaptation analysis employed by Doeller and colleagues[9]: instead of modeling the time passed since the last presentation of a trajectory aligned to 60°, we modeled the angular distance from the previous trial. Given electrophysiological evidence that firing fields of grid cells increase in number around goal location, we assumed that the grid signal could be higher in the quadrants containing the goals (Q2 in the blue context, Q3 in the green one) compared to those that are far apart from the goal (Q3 in the blue context, Q2 in the green one). We thus defined, through SPM12, contrast weights to give a positive value of +1 to the grid modulator for the goal quadrant, -1 for the grid modulator of the quadrant far from the goal, and 0 to the remaining 2 quadrants that had a medium distance (Fig. 4f). Group-level analyses were performed using SPM12.

## Statistical tests

The vast majority of analyses have been carried out at the whole-brain level using SPM12, thus implementing built-in one-sided t-tests. Regions of Interest analyses are usually two-sided t-tests except for

where otherwise indicated because of the non-normality of the data distribution (assessed via Shapiro–Wilk test, statistical details in the text).

## Reporting summary

Further information on research design is available in the Nature Portfolio Reporting Summary linked to this article.

## Data availability

Raw data are protected and are not available due to data privacy. Preprocessed data will be made available upon request to the corresponding author. Processed data are available at https://doi.org/10.17605/OSF.IO/QTHVW[63]. Source data are provided with this paper at https://doi.org/10.17605/OSF.IO/QTHVW[63].

## Code availability

The code for the behavioral task and for generating the main figure results is available at https://doi.org/10.17605/OSF.IO/QTHVW[63].

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

## Acknowledgements

This work was supported by the European Research Council (ERC-StG NOAM awarded to R. Bottini and ERC-CoG GEOCOG 724836 awarded to

C.F.D.) and the Max Planck Society. C.F.D.'s research is further supported by the Kavli Foundation, the Jebsen Foundation, Helse Midt Norge and The Research Council of Norway (223262/F50; 197467/F50). R. Bayramova is also supported by the Max Planck School of Cognition. We would like to thank K. Träger for organizational help and J. Lepsien, M. Jochemko, S. Neubert, D. Klank, and M. Hofman for help during fMRI data acquisition. Finally, we would like to thank all members of the Doeller and Bottini labs for fruitful discussions that triggered some useful changes in this study.

## Author contributions

S.V., R.Ba, C.F.D., and R.Bo conceived the study. S.V. programmed the tasks and collected the data with support from R.Ba and S.V. analyzed the data. S.V. wrote the paper with input from R.Ba, C.F.D., and R.Bo. C.D. and R.Bo acquired the funding.

## Competing interests

The authors declare no competing interests.
