## [Peer Review File · Nature Communications]

Mental search of concepts is supported by egocentric vector representations and restructured grid mapsREVIEWER COMMENTS

Reviewer #1 (Remarks to the Author):

This study investigates the possibility that the mental representation of concepts exploits systems of representation analogous to those used in spatial cognition permitting problem solving by goal-directed navigation within the conceptual space. Drawing on methods derived from spatial cognition and refined in previous studies by this group and others, the authors find compelling initial evidence of neural signatures of this spatial mechanism, identifying some of its properties. These include use of an "egocentric" reference frame (within which target concepts, goals, may be sought) and a grid-like "allocentric" representation whose structure is modulated by the presence of these well-learned goals. This is a rich, substantial and elegantly-designed study with intriguing results, methodological and analytic innovations, and important general implications for human neuroscience and psychology.

Participants are first trained in a task where they systematically manipulate the structure of objects (molecules) in an artificial conceptual space in order to achieve one of two goals (i.e., two target molecules that are fixed across the study). Having achieved a criterion level of performance in training sessions spread over two days, participants are then scanned while being tested on a recall task in which they extrapolate a given manipulation in imagination before being asked to determine whether an specified goal will be reached. On a subset of trials participants are asked about an unpredictable, arbitrary configuration. fMRI data from the period during which they imagine the configurational change is analyzed to reveal the neural processes underpinning the mental search process.

The goals configurations and manipulations required to achieve them are cleverly planned so that they lie in a 2D conceptual space, to get from a starting location to the target configuration, the participant must understand where the target is in this space and effect a trajectory in this space by systematically altering the molecules. To do this efficiently they must choose manipulations that correspond to specific directions in the space, requiring an "egocentric" representation. fMRI adaptation is seen in trials where the same direction in conceptual space is required to reach the goal, with the degree of adaptation being correlated with task performance. Multivariate pattern analysis using a classifier demonstrates that patterns of response associated with one context (goal) can be used to recover the conceptual directions used in the other context. This cross-context decoding reveals a network of regions including medial-parental/cingulate/retrosplenial areas previously implicated in egocentric spatial representation and ego-allo transformations, as well as medial prefrontal areas. Overall the evidence for an egocentric-like representation of the conceptual space is quite compelling and coherent both with previous work on conceptual representation and with the literature on spatial cognition.

The authors carry out additional analyses that suggest that navigation of the conceptual space may be achieved through a process akin to mental rotation - that is, in some regions (inferior parietal lobe and precuneus) patterns of activity are more similar across contexts where the conceptual space is rotated by 180 degrees between contexts. They also find evidence that "allocentric" directions in conceptual space show the six-fold symmetry characteristic of grid representations in entorhinal cortex, medial prefrontal cortex and superior parietal lobule. A separate analysis indicates that patterns elicited by grid-aligned directions show greater repetition suppression in those areas of the conceptual space which contain well-learned goals. The evidence concerning allocentric-like representations adds to and is broadly consistent with previous findings on the topic, and in conjunction with the task provide new evidence of the underlying mechanism through which such allocentric representations are modulated by behavioural relevance and can be used to inform goal-related decisions.

Overall this study represents a very substantial contribution in terms of its findings and its methodological and analytical innovation, none of which would have been possible without the very clever task design. In my judgement the results some analyses are of a more exploratory character as befits such a novel study, and the authors have rightly gone the extra mile to squeeze every drop of

meaning from the data. For me the most compelling results concern the egocentric representation and search process. The mental rotation and allocentric analyses are also important as they provide the first hints as to how these can be linked to a form of long-term memory that is independent of the current task demands.

I have a few minor comments and queries below:

1) the eye-movement analysis seems less compelling in that the correlations, while statistically significant are very small indeed (i.e., they must account for a very small proportion of variance). This might be expected for such an indirect and speculative effect. For what it's worth, I accept that the analysis tends to provide additional support for the "egocentric" interpretation, but I don't think the evidence adds very much to picture established through the fMRI analysis, and it is perhaps a bit of a distraction from the main results, so in my view it might be better included in the already extensive supplementary information, with just a brief mention in the main text. This would also allow the authors to remove Figure 2 i-l, which would go a long way to simplifying this critical but currently over-complex figure in my view.

2) p10 In general the paper is impressively clearly written but the description of the mental rotation logic was somewhat harder to follow, and I believe it could be clarified with edits around lines 356-363 on p10.

3) on p12 after reference 6 on line 416 it would be good to have a new paragraph.

4) on line 422 the reference format should probably be consistent with the others in the article and with that used above for reference 6.

5) p15 line 503 - consider inserting additional caveats around the eye-movement claims. "Intriguingly there were some signs that..." or similar.

6) p15 line 499-500 the English in the sentence beginning "In support to..." is a bit mangled and this should be rephrased for clarity.

7) p15 line 522 "These regions represented", is probably a bit strong so it should say "These regions appear to represent".

8) p16 line 541 "In full support..." This sentence could be toned down a little in line with my earlier comments with regard to the eye-movement findings.

9) p16 line 579 "on the contrary," should probably read "inversely,"

10) p17 line 582 cite e.g., Bicanski and Burgess (2018) and/or Byrne, Becker and Burgess (2007) after "vice versa")

11) p17 line 586 "correspondent to" would probably be clearer as "corresponding to".

12) p19 and elsewhere "aspect ratio" is probably not the right term for the manipulated features (i.e, typically used to refer to the overall height versus width of an object or image), so maybe "bond-length ratio", "feature length ratio" or similar would avoid misunderstandings.

13) p19 line 693 - just a suggestion but here where it says "oxygen and hydrogen" it could say "reagents", and this word could perhaps be useful in some other places too if the authors want to avoid drawing too much attention to the details of the cover story.

14) p19 around lines 699-702; I felt that it would have been useful to understand this earlier (in the main body of the paper). It looks as if participants could only adjust the direction of the facing vector in one direction. This would presumably correspond to either clockwise or anticlockwise (and perhaps it would have some subtle effects on the asymmetry of behaviour/judgements on other tasks). However, I think I am right that the orientation of the conceptual space is arbitrary such that (allowing for the different bond-lengths to be represented on different axes and with arbitrary different signs) each participant might have their own sense of clockwise/anticlockwise, left and right, in the conceptual space? If so this also means that (unless I am missing something) the eye movement effects are difficult to predict or interpret. Perhaps the observed effects are related to biases in the allowed rotation in the collect task or spatial biases (such as reading direction) that participants bring to the task/stimuli?

15) It would be good to see an indication of the distribution of allocentric start positions and allocentric probe locations used in the recall task (in supplementary information).

16) p 23 line 867 "higher similarity with movement": add "allocentric" before movement

17) supplementary Figure 6. Comment. Significant asymmetries may reflect the operation of strategies influenced by biases mentioned in point 14)?

18) supplementary Figure 10. The results in panel a seem important and are perhaps worthy of mention in the body of the paper. It's not quite clear from the caption how the distance regressor was constructed - i.e., is it based on the raw distance or does it include some kind of adaptation? In any event the pattern of activation itself with prominent involvement of hippocampus. In any event, the results look meaningful and likely complement the main results, shedding additional light on ego-allo mechanisms involved in the task.

19) supplementary Figure 13. The caption could be rephrased more clearly where the relationship to previous Doeller and Shine studies is mentioned. Presumably adaptation of visual areas is due to greater low-level visual similarity of stimuli in "allocentrically matched" conditions?

Reviewer #2 (Remarks to the Author):

Viganò and colleagues provide here an interesting take on the concept of mental search, however I am not sure that their data support their claims. The authors designed an original task of visual mental manipulation for which participants under fMRI needed to visualize, recognize, and retrieve conceptual goals in a conceptual 2D feature spaces while imagining the drawing of two target molecules. Based on their results, the authors propose that (1) the medial parietal cortex supports an egocentric-like representation of conceptual spaces, that this space can be (2) mentally rotated and (3) is adjoint to grid-like representations in the entorhinal cortex, the medial prefrontal cortex, and the superior parietal lobule – representations that present goal-induced alterations of the grid towards the goal location, in a similar fashion to what has been observed in rodent cognitive map.

While the manuscript is well written and flows nicely, I have some major concerns related to the conceptualization of the task and the interpretations of the results – see below.

Major concerns:

The designed task is interesting, however the way the authors chose to analyze and interpret it

presents, in my opinion, several challenges:

1. Most problematical for me is the fact that the authors decided to not consider the actual 2D space that the participants are mentally working in, but rather a conceptual 2D space where "x" is the length of one of the bonds of the molecule and "y" the length of another bond. This creates some confusion as to the interpretation of the results such as eye movements (fig 3) or grid-like representation (fig 4).

– I would suggest for the authors to first analyze all their result in the "real" 2D space before investigating the conceptual space. I think that their results would have been much stronger when comparing those two spaces explicitly.

2. The classical definition of egocentric navigation refers to using oneself perspective as a frame of reference (turning left/right in respect to ones' position – egocentric vectors) as opposed to allocentric navigation for which one uses stable landmarks in the external space to create an allocentric map. If I understood correctly, the authors suggest that in their task, egocentric navigation refer to the mental process by which participants visualize the atom of a molecule from where it is to where it would be if an atom-atom bond continue to be morphed at a given rate – as if they were placing themselves in the position of the molecule. While I acknowledge that previous reports (including work from the authors) have used the term "egocentric" to describe similar notions, I am not convinced that this is the right concept.

– This is problematic given that the whole premises of this study is based on this concept. I would suggest for the authors to – at least – discuss clearly how broad and not classical is their use of the term "egocentric", as well as better justify why they consider that egocentric navigation is the right denomination of the phenomenon they are testing – this both in the introduction and the discussion.

– The use of a conceptual space further complicates the notion of egocentric vectors given that – most likely – participant do not explicitly use such a conceptual space. I am therefore unsure whether the authors results support their claims. Please justify.

3. About mental rotation – First, while this task requires 2D spatial mental visualization, I am not sure to which extend it requires mental rotation. I am therefore wondering whether this task is the best to test mental rotation. This should be discussed. Second, the two goal molecules are symmetrical which lead to some issues when it comes to mental rotation. I acknowledge that the authors partially controlled for these caveats in fig 3. It is however in my opinion insufficient, given that the authors fail to take into consideration the fact that non-goal bonds are also symmetrical. Please control for this.

Minor concerns:

1. I noticed a couple of sentences that I think should be slightly reformulated in their assumption to what is unique to human cognition:

– L 39 One of the hallmarks of our species – I am not convinced that this is a hallmark of human vs. other mammals.

– L53 "the brain circuits that evolved for navigating physical spaces in other mammals might be used in our species for organizing conceptual knowledge, enabling us to "mentally navigate" through concepts and memories as if they were locations in our internal conceptual spaces» – we do not know it is not the case in other mammals.

– L55 not only human navigation goes beyond hippocampus cognitive map.

2. Reference to egocentric navigation studies in rodents are missing: e.g.: object vector cells (Moser), egocentric cells (Knierim, Hasselmo, Derdikman). Please cite this work and contextualize your findings given what we know from those studies.

3. I think that it would be interesting to provide a video of the visual task in the supplementary material.

4. sf3 b: Why directions are represented by black bars on the left and blue wedges on the right?
5. It would be interesting to explore how representations may change in individuals with bad performance. Please test or refer to the figure where it is done (in case I missed it).
6. fMRI adaptation analyses: given the generalist readership of nature communication, I would suggest to better explain the rationale behind this analysis and refer to original studies (Grill-Spector, Krekelberg), as well as discuss the specificity of the task, including how the time lapse used to assess fMRI suppression is adequate here.
7. Same comments with MVPA: Please spell out (multivariate pattern analysis), explain rationale behind analysis, provide references to original and discuss differential nature of results as compared to adaptation analyses. – As a sign note: I find it very interesting that the authors are using two types of analyses here. I think it would be even better if they were taking full advantage of this by explaining how these methods differs and may be complementary (taking in consideration their respective caveats).
8. sf 9 what is a and b? Blue vs. green context? Please adjust legend. The imaging results do not seem clear-cut as to the zones of activation – please discuss.
9. For all fMRI control (e.g., sf 10 – but not restricted to that figure), I would suggest adding an overlay of area of activation in original conditions (with a dashed contour for example). This would allow the reader to have better visualization of how original and controls signals – for example in the medial occipito-parietal cortex and precuneus – may overlap.
- 10: Most of the analyses are based on correlations. I would be much more convinced by decoding/predictive analyses. Could the authors attempt to decode performance base on activation zone for example?
11. I am very unconvinced as to the interpretation of the eye tracking experiments. See major concern. Please compare analyses between “real” 2D space and conceptual space. Please justify better why eye tracking can apply to a conceptual space.
12. Grid-like signal: it is not clear which structures are analyzed: Only mEC and IEC?

Reviewer #3 (Remarks to the Author):

The manuscript by Vigano and colleagues describes a human fMRI study in which subjects learned the locations of 2 different stimuli each in a 2-dimensional conceptual space. Subjects were required to “navigate” to these goals in order to perform a behavioral task. While the 2-d conceptual space was not made explicit to subjects, they nonetheless performed well on the task with their behavioral responses and even their eye movements suggesting that they did represent the 2-dimensional space and did so from an egocentric perspective. fMRI analyses (using adaptation and multivoxel pattern analyses) provided converging support for an egocentric coding of space, particularly in medial parietal cortex, that reflected the relative orientation of subjects as they navigated the space on each trial. Separately, entorhinal cortex (and other areas) showed a grid-like allocentric representation of space (at least based on fMRI adaptation measures) that was relatively stronger when subjects were close to the goal location.

The manuscript addresses a very interesting question about whether humans form and can navigate conceptual spaces in an egocentric reference frame. The analyses are well motivated by prior work and complement prior findings. However, the current findings do represent a significant advance on prior work by considering the egocentric navigation. The experimental paradigm is clever and the results include a commendable number of converging analyses. While I think some of the analyses are a bit opaque/dense, I was able to follow the logic and did not see major problems in their implementation or interpretation. Overall, I found this a very interesting paper and believe it will likely be of broad interest. I only have a few comments/questions.

1. Could the egocentric reference frame simply reflect whether subjects were getting closer or farther to the goal? From the analyses reported, it's hard to know whether the egocentric representation was fine-grained (sensitive to minor changes in angle) or very coarse (e.g., getting closer or farther). And with the 45 degree increments, maybe this is not addressable. But do the authors think the representation was more fine-grained than closer/farther? If so, is there any aspect of the data that would support this?

2. Was the distribution of angles to the target fully independent of the starting position? Of particular concern, was there any correlation such that certain angles tended to be closer/farther from the goal? The distance-controlled analysis suggests this might not be a problem, but it also made me wonder if this was a confound in the design (and that was the motivation for the distance-controlled analysis)?

3. Similarly, it would be informative to know if the parietal regions code the distance to the goal. It would be striking, for example, if the parietal regions do not code distance to goal, but do code for angle. This would make for an even stronger argument about egocentric navigation in conceptual space.

4. Were subjects debriefed in any way to gather information about their strategy for performing the task? Although they were not explicitly made aware of the 2d structure, did some/any of the subjects figure this out on their own?

Replies from the Authors are in blue and bold font.

REVIEWER COMMENTS

Reviewer #1 (Remarks to the Author):

This study investigates the possibility that the mental representation of concepts exploits systems of representation analogous to those used in spatial cognition permitting problem solving by goal-directed navigation within the conceptual space. Drawing on methods derived from spatial cognition and refined in previous studies by this group and others, the authors find compelling initial evidence of neural signatures of this spatial mechanism, identifying some of its properties. These include use of an "egocentric" reference frame (within which target concepts, goals, may be sought) and a grid-like "allocentric" representation whose structure is modulated by the presence of these well-learned goals. This is a rich, substantial and elegantly-designed study with intriguing results, methodological and analytic innovations, and important general implications for human neuroscience and psychology.

Participants are first trained in a task where they systematically manipulate the structure of objects (molecules) in an artificial conceptual space in order to achieve one of two goals (i.e., two target molecules that are fixed across the study). Having achieved a criterion level of performance in training sessions spread over two days, participants are then scanned while being tested on a recall task in which they extrapolate a given manipulation in imagination before being asked to determine whether an specified goal will be reached. On a subset of trials participants are asked about an unpredictable, arbitrary configuration. fMRI data from the period during which they imagine the configural change is analyzed to reveal the neural processes underpinning the mental search process.

The goals configurations and manipulations required to achieve them are cleverly planned so that they lie in a 2D conceptual space, to get from a starting location to the target configuration, the participant must understand where the target is in this space and effect a trajectory in this space by systematically altering the molecules. To do this efficiently they must choose manipulations that correspond to specific directions in the space, requiring an "egocentric" representation. fMRI adaptation is seen in trials where the same direction in conceptual space is required to reach the goal, with the degree of adaptation being correlated with task performance. Multivariate pattern analysis using a classifier demonstrates that patterns of response associated with one context (goal) can be used to recover the conceptual directions used in the other context. This cross-context decoding reveals a network of regions including medial-parental/cingulate/retrosplenial areas previously implicated in egocentric spatial representation and ego-allo transformations, as well as medial prefrontal areas. Overall the evidence for an egocentric-like representation of the conceptual space is quite compelling and coherent both with previous work on conceptual representation and with the literature on spatial cognition.

The authors carry out additional analyses that suggest that navigation of the conceptual space may be achieved through a process akin to mental rotation - that is, in some regions (inferior parietal lobe and precuneus) patterns of activity are more similar across contexts where the conceptual space is rotated by 180 degrees between contexts. They also find

evidence that "allocentric" directions in conceptual space show the six-fold symmetry characteristic of grid representations in entorhinal cortex, medial prefrontal cortex and superior parietal lobule. A separate analysis indicates that patterns elicited by grid-aligned directions show greater repetition suppression in those areas of the conceptual space which contain well-learned goals. The evidence concerning allocentric-like representations adds to and is broadly consistent with previous findings on the topic, and in conjunction with the task provide new evidence of the underlying mechanism through which such allocentric representations are modulated by behavioural relevance and can be used to inform goal-related decisions.

Overall this study represents a very substantial contribution in terms of its findings and its methodological and analytical innovation, none of which would have been possible without the very clever task design. In my judgement the results some analyses are of a more exploratory character as befits such a novel study, and the authors have rightly gone the extra mile to squeeze every drop of meaning from the data. For me the most compelling results concern the egocentric representation and search process. The mental rotation and allocentric analyses are also important as they provide the first hints as to how these can be linked to a form of long-term memory that is independent of the current task demands.

We thank the Reviewer for the careful analysis of our study and for the very positive feedback. Below are listed our point-by-point replies to the comments.

I have a few minor comments and queries below:

1) the eye-movement analysis seems less compelling in that the correlations, while statistically significant are very small indeed (i.e., they must account for a very small proportion of variance). This might be expected for such an indirect and speculative effect. For what it's worth, I accept that the analysis tends to provide additional support for the "egocentric" interpretation, but I don't think the evidence adds very much to picture established through the fMRI analysis, and it is perhaps a bit of a distraction from the main results, so in my view it might be better included in the already extensive supplementary information, with just a brief mention in the main text. This would also allow the authors to remove Figure 2 i-l, which would go a long way to simplifying this critical but currently over-complex figure in my view.

We agree with the Reviewer that the results from the eye movement analysis with DeepMR eye is statistically less compelling than the neuroimaging results (although significant), however we believe that they provide evidence for an important and decisive conceptual point, namely the recruitment of a 1-st person perspective during goal-directed mental search in conceptual space. We would like to highlight a few aspects which would justify the inclusion of the results in the manuscript:

- **firstly, the validation procedure performed with DeepMR eye in our study: as explained in the text, we did not analyze immediately our data of interest (imagination period), rather we first validated the ability of the analytical approach to distinguish between left vs right using an independent partition of the data, that is the period of answer choice (Yes vs No, or No vs Yes). Only after having checked for the validity of the approach, we applied the same analysis to the period of interest.**

- secondly, we implemented a control for our main analysis of interest about the left vs right effect, namely we verified that no correlation with eye movements should be seen in the vertical dimension. This control is important, as the stimuli that participants are seeing actually vary along the vertical axis during the morphing period. This analysis demonstrates that eye movements are not correlated with this dimension but rather with the horizontal one, which doesn't really carry information on the "real screen/space" but putatively conveys all the information in the conceptual space. This is of particular relevance in our opinion;
- finally, we showed that there was no difference between the two contexts, with the effect going in the (expected) direction for both: this further supports the idea that the two conceptual environments are potentially realigned to maintain the same egocentric point-of-view to the goal.

Overall, we think that this result is of interest and the first of its kind in this research line, therefore we feel that readers would benefit from having it available in the main text. We are clearly open to considering moving these results into the supplements in case the Reviewer would strongly recommend so. We also have toned down the interpretation of the eye movement results when discussing the results in the revised manuscript, see below.

2) p10 In general the paper is impressively clearly written but the description of the mental rotation logic was somewhat harder to follow, and I believe it could be clarified with edits around lines 356-363 on p10.

We apologize with the Reviewer for the lack of clarity. The paragraph has been corrected and extended in the following way:

"If this hypothesis is correct, a specific prediction follows: the representations of the two contexts or environments (blue vs green molecule) should be aligned, in such a way that the goal quadrant from one conceptual space should be more similar to the goal quadrant in the other conceptual space and, crucially, the non-goal quadrants from the two environments would also correspond across contexts following their relative position to the goal. As an example, consider an observer at the center of the green environment facing the goal quadrant (Q3) in Fig. 3a. According to her point of view, Q1 would be to the left of the goal, and Q4 would be to the right. Now consider the same observer in the blue context, facing the context-relevant goal-quadrant (Q2). In this case, Q4 would be to the left, and Q1 to the right. If participants are maintaining the focus of attention from their own current state to the goal state, then we might assume that not only Q3 and Q2 in the green and blue context respectively should be represented similarly (they are the goal quadrants "in front of the observer"), but also that the rest of the quadrants would be represented in the same relationship to each other across the two contexts. This can be seen as an example of "alignment" along the common left-right axis, which is orthogonal to sagittal plane created by connecting our current state with the goal state and actually dividing the space into left and right (see also ref. 22 for visual representations of how this works for spatial navigation in rodents). How can we test this?"

First, we extracted activity patterns for the 4 quadrants separately for the two contexts. Then we looked, using a whole-brain searchlight, for brain regions that had an increased similarity between the quadrants in one context (e.g., blue molecule) with the corresponding quadrants in the other one (e.g. green molecule) after the feature space was rotated by 180°. In other words, after we aligned the goal locations in both contexts according to the model just described (e.g., the activity evoked by trials occurring in quadrant 1 (Q1) for the blue molecule would be more similar to that evoked by trials in Q4 for the green molecule, see Methods and Fig. 3a).”

3) on p12 after reference 6 on line 416 it would be good to have a new paragraph.

We agree with the Reviewer that this will increase the readability of the Results Section, therefore we have now splitted the longer paragraph into two shorter ones.

4) on line 422 the reference format should probably be consistent with the others in the article and with that used above for reference 6.

Corrected, thanks

5) p15 line 503 - consider inserting additional caveats around the eye-movement claims. "Intriguingly there were some signs that..." or similar.

We have now added the sentence “First, we showed evidence of egocentric-like codes in the parietal cortex to represent goal position in conceptual spaces relative to our current state or experience. Intriguingly, we also observe preliminary evidence for egocentric representations reflected in the pattern of eye movements”.

6) p15 line 499-500 the English in the sentence beginning "In support to..." is a bit mangled and this should be rephrased for clarity.

We have now modified it with “In line with this hypothesis, in the current study we reported three main novel findings, showing the parallel recruitment of complementary reference frames to track conceptual goals during mental search.”

7) p15 line 522 "These regions represented", is probably a bit strong so it should say "These regions appear to represent".

Corrected

8) p16 line 541 "In full support..." This sentence could be toned down a little in line with my earlier comments with regard to the eye-movement findings.

We have toned down the whole sentence in the following way: “Supporting this, the additional, preliminary, evidence coming from the analysis of eye movements using DeepMRye suggested that participants were employing a first-person perspective of the conceptual spaces while mentally searching through them.”

9) p16 line 579 "on the contrary," should probably read "inversely,"

Corrected

10) p17 line 582 cite e.g., Bicanski and Burgess (2018) and/or Byrne, Becker and Burgess (2007) after "vice versa")

Done

11) p17 line 586 "correspondent to" would probably be clearer as "corresponding to".

Corrected

12) p19 and elsewhere "aspect ratio" is probably not the right term for the manipulated features (i.e, typically used to refer to the overall height versus width of an object or image), so maybe "bond-length ratio", "feature length ratio" or similar would avoid misunderstandings.

Corrected throughout the whole manuscript (not highlighted in blue)

13) p19 line 693 - just a suggestion but here where it says "oxygen and hydrogen" it could say "reagents", and this word could perhaps be useful in some other places too if the authors want to avoid drawing too much attention to the details of the cover story.

We thank the Reviewer for the suggestion and we have now changed some instances of this in the manuscript

14) p19 around lines 699-702; I felt that it would have been useful to understand this earlier (in the main body of the paper). It looks as if participants could only adjust the direction of the facing vector in one direction. This would presumably correspond to either clockwise or anticlockwise (and perhaps it would have some subtle effects on the asymmetry of behaviour/judgements on other tasks). However, I think I am right that the orientation of the conceptual space is arbitrary such that (allowing for the different bond-lengths to be represented on different axes and with arbitrary different signs) each participant might have their own sense of clockwise/anticlockwise, left and right, in the conceptual space? If so this also means that (unless I am missing something) the eye movement effects are difficult to predict or interpret. Perhaps the observed effects are related to biases in the allowed rotation in the collect task or spatial biases (such as reading direction) that participants bring to the task/stimuli?

We apologize for the lack of clarity and for missing to report an important detail. Participants could adjust the direction of the facing vector in both directions, by pressing either Z or Y (German keyboard). The procedure has been inspired by a previous paper by Constantinescu and colleagues (2016 Science). We have now corrected the manuscript accordingly. In short, participants could cover the full spectrum of movements: rotation clockwise and counterclockwise (Z and Y), as well front or back (C and V, when applying the change/movement). Please also refer to the example video of the task that we made available.

Having said this, we recognize that the Reviewer is right, and each participant may have his/her own sense of clockwise/anticlockwise. Indeed, a priori, we could not make strong predictions about a specific sense and also for that reason, maybe, the eye movement results should be taken as preliminary - as we now stress in the paper. On the other hand, however, the Reviewer is right in thinking that subjects may arrive with some common bias that could explain a common configuration of the conceptual space that can be reflected in eye movements. However, we do not think that the bias may derive from particular mechanical constraints imposed during the training (see first part of this response). Instead, we consider more likely that the clockwise (counter-clockwise) movements in terms of egocentric bearing from a reference vector (the goal vector) correspond to rightward (leftward) movements as it would naturally happen if the goal vector reflected the subject's facing direction. Also for this reason we think that the eye movement results are important to provide a full picture of the processes we are investigating in the study.

15) It would be good to see an indication of the distribution of allocentric start positions and allocentric probe locations used in the recall task (in supplementary information).

We have added an image representing the starting (and ending) positions for one subject in the Supplementary Figure 3. Please notice that the starting coordinates are mostly clustered in the central region of each quadrant because in this way we could make sure that morphing trajectories would not cross quadrant borders, as mentioned in the Methods, thus allowing us to more properly test for modulations of the grid-like signal as a function of distance from the goal (here operationalised as a function of the quadrant). At the same time, this approach controlled for influence of additional potential confounding factors such as distance from borders, which would require a more dedicated experimental design. Finally, close starting points diminish the influence of small variations of allocentric starting positions across egocentric conditions of interest: although this was anyway controlled for (see Supplementary Materials), we reasoned that minimizing this effect from the beginning would have been beneficial for isolating the angular effect of interest.

16) p 23 line 867 "higher similarity with movement": add "allocentric" before movement

Corrected

17) supplementary Figure 6. Comment. Significant asymmetries may reflect the operation of strategies influenced by biases mentioned in point 14)?

Following our reply to point 14, we think the two aspects are unlikely to be related. We agree with the Reviewer that these asymmetries are interesting, and we are investigating this in follow-up studies.

18) supplementary Figure 10. The results in panel a seem important and are perhaps worthy of mention in the body of the paper. It's not quite clear from the caption how the distance regressor was constructed - i.e., is it based on the raw distance or does it include some kind of adaptation? In any event the pattern of activation itself with prominent involvement of

hippocampus. In any event, the results look meaningful and likely complement the main results, shedding additional light on ego-allo mechanisms involved in the task.

The regressor modeled whether the morphing molecule was getting closer or further apart to the goal, by taking the Euclidean distance between allocentric end position and allocentric start position as a proxy of this. We agree with the Reviewer that this is a relevant finding and we have now included a short reference in the Results section of the main text.

19) supplementary Figure 13. The caption could be rephrased more clearly where the relationship to previous Doeller and Shine studies is mentioned. Presumably adaptation of visual areas is due to greater low-level visual similarity of stimuli in "allocentrically matched" conditions?

We have now corrected the caption with: "The resulting clusters reveal similarities with previous studies showing allocentric head-direction like signals in the human brain during virtual reality fMRI experiment e.g., Doeller et al. 2010 and Shine et al. 2015, specially in the RSC and the Thalamus (Tha)."
Regarding the adaptation in visual areas, yes, this is a likely interpretation.

Reviewer #2 (Remarks to the Author):

Viganò and colleagues provide here an interesting take on the concept of mental search, however I am not sure that their data support their claims. The authors designed an original task of visual mental manipulation for which participants under fMRI needed to visualize, recognize, and retrieve conceptual goals in a conceptual 2D feature spaces while imagining the drawing of two target molecules. Based on their results, the authors propose that (1) the medial parietal cortex supports an egocentric-like representation of conceptual spaces, that this space can be (2) mentally rotated and (3) is adjoint to grid-like representations in the entorhinal cortex, the medial prefrontal cortex, and the superior parietal lobule – representations that present goal-induced alterations of the grid towards the goal location, in a similar fashion to what has been observed in rodent cognitive map.

While the manuscript is well written and flows nicely, I have some major concerns related to the conceptualization of the task and the interpretations of the results – see below.

We thank the Reviewer for the interest in our manuscript and for the constructive comments. Our point-by-point replies are listed below, and we hope they will convince the Referee about the validity of our results and interpretations.

Major concerns:

The designed task is interesting, however the way the authors chose to analyze and interpret it presents, in my opinion, several challenges:

1. Most problematical for me is the fact that the authors decided to not consider the actual 2D space that the participants are mentally working in, but rather a conceptual 2D space where “x” is the length of one of the bonds of the molecule and “y” the length of another bond. This creates some confusion as to the interpretation of the results such as eye movements (fig 3) or grid-like representation (fig 4).

I would suggest for the authors to first analyze all their result in the “real” 2D space before investigating the conceptual space. I think that their results would have been much stronger when comparing those two spaces explicitly.

We apologize for any unclarity in the manuscript which might have led to a misunderstanding of our task. We are not sure what the Reviewer refers to as the “actual 2D space that the participants are mentally working in”. In our experiment, participants are laying in the MRI scanner and are looking at a screen monitor where we projected visual shapes (the molecules). These stimuli, being differentiated only by changes in their bond-length ratio, are conceivable as points in a 2D feature space defined by the length of these two bonds. Therefore, the 2D space that participants are mentally working in is exactly what we refer to as conceptual space (in our case, a feature space where salient regions/exemplars are named and have some semantic attributes as in our cover story). Therefore, there exist only two spaces that can matter for the participant:

- 1. the real visual space that they see in front of their eyes, defined by the bidimensional screen and what happens on it;**

2. the conceptual space of the molecules wherein participants' "mental effort" is happening (otherwise they couldn't solve the task).

Thus, we assume that the Reviewer refers to the first type of space, when he or she talks about the "real 2D space", perhaps having in mind studies like Julian et al. 2018 Nat Neuro or Nau et al. 2018 Nat Neuro. In our study, however, there is nothing happening on the screen that could represent a confound (our design, indeed, is more similar to the one of Constantinescu et al. 2016 Science): the molecule is always presented centrally, with the upper and lower bonds stretching vertically and the only relevant event happening is their ratio of change. But this ratio already requires, at the mental level, the integration (or if you prefer, projection) of the stimulus on a 2D internal abstract space, for the task to be solved. In other words, in our case (as in the Constantinescu et al study), there is no 2-dimensional displacement on the screen nor in visual space, since the molecules morph along a vertical axis only. The only way to obtain 2D trajectories describing different angles (egocentric or allocentric) is by considering each molecule configuration in the underlying 2D feature space defined by the length of the bonds. This conceptual space is akin to the cognitive map formed by the participants in order to solve the task. The only potential confounding factor that we can think of, prompted by the Reviewer's comment, is the fact that molecules are changing along the vertical axis. This, however, is controlled for by the analyses that we have already reported: the eye tracking results indeed show that the relevant information is captured by horizontal eye movements significantly more than vertical, and that the latter, on the contrary, doesn't show significant effects. Given that on the screen there is no change happening horizontally in front of the participant during morphing or imagination (e.g., the molecule is not moving around, nor it is changing in any way horizontally), we can conclude that eye movements are reflecting the types of mental processes happening inside participants' minds.

In short: the only real space that participants are exposed to is the visual screen used to project the stimuli, on which molecules are always presented centrally and they only show minimal stretching along the vertical axis (see video). To solve the task, we believe that it is very unlikely to interpret these changes in any other way than that participants build up a mental representation of the underlying concept space, for which we indeed report consistent neuroimaging results. The analysis of eye movements further confirms this by showing that horizontal gaze displacement correlates with the predicted change in mental space, while no horizontal information is presented on the visual screen. Similar arguments apply to the results of the hexadirectional analysis, where the reported 6-fold periodicity (and its modulation) can only refer to mental space and representations, as there are no moving molecules or visually displayed directions on the screen (in contrast to for instance with Julian et al. and Nau et al. 2018's papers).

We apologize to the Reviewer if any part of our manuscript was not clear enough, and we are open to suggestions on what to describe better to avoid this confusion in the future. We also want to stress that our current design doesn't create a situation in which two competing sets of information, one visually presented in the physical world, and one relevant in the mental space, have to be simultaneously evaluated: this would require a specific experimental design where precise manipulations of

both the physical and the conceptual environment coexist and lead to different predictions. This was not the case of our experiment, where the visual space was minimally informative.

2. The classical definition of egocentric navigation refers to using oneself perspective as a frame of reference (turning left/right in respect to ones' position – egocentric vectors) as opposed to allocentric navigation for which one uses stable landmarks in the external space to create an allocentric map. If I understood correctly, the authors suggest that in their task, egocentric navigation refer to the mental process by which participants visualize the atom of a molecule from where it is to where it would be if an atom-atom bond continue to be morphed at a given rate – as if they were placing themselves in the position of the molecule. While I acknowledge that previous reports (including work from the authors) have used the term “egocentric” to describe similar notions, I am not convinced that this is the right concept.

This is problematic given that the whole premises of this study is based on this concept. I would suggest for the authors to – at least – discuss clearly how broad and not classical is their use of the term “egocentric”, as well as better justify why they consider that egocentric navigation is the right denomination of the phenomenon they are testing – this both in the introduction and the discussion.

The use of a conceptual space further complicates the notion of egocentric vectors given that – most likely – participant do not explicitly use such a conceptual space. I am therefore unsure whether the authors results support their claims. Please justify.

While we acknowledge that physical navigation is different from solving our concept task, we believe that our experimental design and results actually fit very well the definition given by the Reviewer. Indeed, as reported on p. 5 of the manuscript, goal location was represented in parietal cortex independently from the allocentric position of the goal (which was different across the two contexts), or from the allocentric angle described by the trajectory, or from the current allocentric position in the conceptual space. Goal location in parietal cortex was defined exclusively on the basis of the relative position of the goal relative to the observer (X degrees on one side or the other) with respect to the current direction in the conceptual space - this is akin to encoding whether an object (goal) is X degrees to the left or to the right (i.e., on one side or the other) with respect to a current facing direction. Geometrically, this is a viewpoint-dependent code in which the viewpoint is defined by the current morphing trajectory as, in the real space, would be defined by the current facing direction. Indeed, according to human fMRI studies, this neural code emerges in a medial-parietal region that has been shown to encode egocentric (viewpoint-dependent) goal direction during spatial navigation (Chadwick et al. 2015). In other words, all our analysis (adaptation, cross decoding, eye-movements, mental rotation) converge in showing that, in parietal cortex, goal objects that are fixed in allocentric terms are represented differently based on the specific “egocentric-like” vectors that underlie each trial trajectory.

The relevant passages in the main text are the following:

P 5: “Critically, the trajectories implied by the morphing molecules could be directed to the goal (0° from correct trajectory; on-target trials, participants are expected to respond “Yes”) or deviating from it by a certain degree (-135°, -90°, -45°, +45°, +90° +135° from correct trajectory; off-target trials, the expected response was “No”. Signs could be interpreted as arbitrary “left” and “right”, see Methods and Supplementary Figure 3): these angular conditions will be referred to as “egocentric-like conditions”, because they indicate where the goal is in the feature space with respect to the current trajectory, independently from their allocentric position and from the angle of movement relative to the general layout of the feature space (Fig. 2a). “

P 16: “In our study, we reported remarkably similar observations in the parieto-occipital sulcus (POS), the precuneus, and the PPC extending to the RSC. These regions appeared to represent goal locations in conceptual spaces according to a reference frame that was independent from the position of the goal across contexts, the allocentric angle of the trajectory, and the current allocentric position in space. This representational scheme can be interpreted as an egocentric vector relative to the goal: Akin to the physical world, where we can tell whether an object is to our left or right while we move and change position in the environment, here, by considering how the bond-length ratio of the molecule stimuli changed, we could define to what degree a goal concept was on one side or the other in the underlying feature space, irrespective to its external geometry. This indicates the recruitment of egocentric-like schemes in the parietal cortex for representing conceptual goals.”

We have now added this additional paragraph immediately after the first one to explain the rationale of our design in more detail and we hope that this will help to answer the Referee’s question

“In other words, as it happens in the physical environment where, given an oriented agent (“viewpoint”) positioned in it, and a specific goal or location to refer to (the “target”), we can define whether the target is positioned to one side (“left”) or the other (“right”) of our current point of view, here we had an environment (the molecule space), a target (the goal molecule), and a point of view (defined by the morphing direction of the molecule) that allowed us to mimic the “one side vs the other side” (left vs right) distinction.”

3. About mental rotation – First, while this task requires 2D spatial mental visualization, I am not sure to which extent it requires mental rotation. I am therefore wondering whether this task is the best to test mental rotation. This should be discussed. Second, the two goal molecules are symmetrical which lead to some issues when it comes to mental rotation. I acknowledge that the authors partially controlled for these caveats in fig 3. It is however in my opinion insufficient, given that the authors fail to take into consideration the fact that non-goal bonds are also symmetrical. Please control for this.

The experiment was not designed to test for mental rotation, but given the results of our cross-context analyses, we tested whether the reported results were compatible with a representational code wherein corresponding portions of the two environments were aligned to each other. Our key interpretation of this effect is contextual realignment but this representation is also reminiscent of mental rotation findings in

the literature. Please notice that in the Discussion we refer to this additional interpretation as still speculative. Nevertheless, the evidence for alignment between the two contexts (which is the key aspect) is, in our opinion, clear and convincing, surviving also crucial controls (see Figure 3).

For what concerns the symmetry of the molecules, we are not sure how this could affect our results or interpretation. We acknowledge, however, that this could have helped the participants in performing this form of mental rotation/alignment, therefore we have now added a reference to this issue in the Discussion:

“Additionally, it should be acknowledged that the process of mental realignment between the two contexts might have been facilitated and made more spontaneous by the symmetry of the visual stimuli used, and new studies should address whether and how competitive visual factors (e.g., asymmetrical bonds in our molecules) affect the results.”

Moreover, to better account for the Reviewer’s suggestion, we have now modified the subtitle of the relevant Results section mentioning “realignment”, which is a more parsimonious term that, we hope, will be in line with the Reviewer’s thoughts.

Minor concerns:

1. I noticed a couple of sentences that I think should be slightly reformulated in their assumption to what is unique to human cognition:

L 39 One of the hallmarks of our species – I am not convinced that this is a hallmark of human vs. other mammals.

We agree with the Reviewer and we have now changed the text: “A crucial characteristic of an intelligent mind” which we think applies more broadly to other species as well

L53 “the brain circuits that evolved for navigating physical spaces in other mammals might be used in our species for organizing conceptual knowledge, enabling us to “mentally navigate” through concepts and memories as if they were locations in our internal conceptual spaces» – we do not know it is not the case in other mammals.

That’s true, in fact we have now added a reference to the important paper by Aronov et al. 2017 Nature.

L55 not only human navigation goes beyond hippocampus cognitive map.

We have now removed the reference to humans

2. Reference to egocentric navigation studies in rodents are missing: e.g.: object vector cells (Moser), egocentric cells (Knierim, Hasselmo, Derdikman). Please cite this work and contextualize your findings given what we know from those studies.

We thank the Reviewer for pointing us to this oversight. We now cite additional evidence of egocentric coding in non-human species.

“[...] *egocentric* representations (usually associated with parietal regions) encode object position with respect to the observer’s perspective, and changes as a function of the observer’s movements ²⁰⁻²² (for similar findings in non-human species see reff ⁵⁹⁻⁶²)”

We have also added a final remark at the end of the Discussion opening to the possibility that similar mechanisms could be isolated in other animals.

“Whether these results extend to non-human species as well (as for instance happens for allocentric map-like coding ⁵⁵) cannot be concluded with the current experiment, but our findings [...]”

3. I think that it would be interesting to provide a video of the visual task in the supplementary material.

Thanks for this suggestion. Videos for both the collect task and the recall task have now been added to the Supplementary Materials.

4. sf3 b: Why directions are represented by black bars on the left and blue wedges on the right?

Directions on the left are egocentric directions to the goal and have precise frequency values (those reported in figure 2a); directions on the right are allocentric directions that span with higher granularity the 0-359° range. Please notice that for our egocentric analyses a smaller number of egocentric directions with multiple repetitions and balanced across the two “sides” of the egocentric perspective is sufficient (and actually preferable, e.g., for decoding), while for the allocentric analysis, a broader sampling of the directions is necessary for a powerful hexadirectional analysis.

5. It would be interesting to explore how representations may change in individuals with bad performance. Please test or refer to the figure where it is done (in case I missed it).

We assume the Reviewer is referring to a possible link between neural representation and performance. Figure 2d shows that “bad” performers have in general lower adaptation effects. This potentially indicates that the representation of the different egocentric conditions are less precise or distinguishable, as if they were “erroneously categorized”. We believe that our sample size, despite consisting of 40 subjects, is still too small for more precise and proper analyses of interindividual differences, but we agree with the Reviewer that this would be a very interesting line of research.

6. fMRI adaptation analyses: given the generalist readership of nature communication, I would suggest to better explain the rationale behind this analysis and refer to original studies

(Grill-Spector, Krekelberg), as well as discuss the specificity of the task, including how the time lapse used to assess fMRI suppression is adequate here.

Thank you. See response to point 7 below.

7. Same comments with MVPA: Please spell out (multivariate pattern analysis), explain rationale behind analysis, provide references to original and discuss differential nature of results as compared to adaptation analyses. – As a sign note: I find it very interesting that the authors are using two types of analyses here. I think it would be even better if they were taking full advantage of this by explaining how these methods differ and may be complementary (taking in consideration their respective caveats).

For both comments 6 and 7 we agree with the Reviewer that some extra details on the used methodology might be useful, therefore we have added more introductory paragraphs on both adaptation and fMRI in the relevant Methods section and a small reference to the advantage of a combined approach in the Results section. Please notice that the word limit imposed by the Journal requires some compromise in discussing these broad methodological aspects.

“The first analytical approach that we implemented was a univariate adaptation analysis, building on the observation that fMRI BOLD signal shows suppression (or adapts) when stimuli are repeated, potentially because of the adaptation of the underlying neuronal populations^{56,57}. We reasoned that if a brain region is representing specific egocentric conditions differently, then it should show a suppression pattern specific to each individual condition.”

“The second analytical approach that we implemented was a multivariate decoding analysis (also known as Multivoxel Pattern Analysis or MPVa⁵⁸): here, instead of analyzing the univariate BOLD change at the single voxel level, the distributed activity pattern in a specified region is considered, and each voxel provides its own contribution for training (and then testing) a multivariate classifier in correctly decoding the presented condition”

8. sf 9 what is a and b? Blue vs. green context? Please adjust legend. The imaging results do not seem clear-cut as to the zones of activation – please discuss.

The Reviewer is correct, they refer to the blue and green molecules. We apologize to the Reviewer for the lack of clarity and we have now corrected the legend. Moreover, we have added the following paragraph to provide a tentative interpretation of the specific differences: “Nevertheless, evident differences were observable in the specific and precise anatomical positions of some of the clusters. A potential interpretation, in light of the mental rotation analysis (see Results and Discussion) is that participants were maintaining one of the two contexts as reference and were rotating, or aligning, the second, thus recruiting additional brain regions. At the present moment this remains an interpretation, which would need further experiments to be properly clarified.”

9. For all fMRI control (e.g., sf 10 – but not restricted to that figure), I would suggest adding an overlay of area of activation in original conditions (with a dashed contour for example). This would allow the reader to have better visualization of how original and controls signals – for example in the medial occipito-parietal cortex and precuneus – may overlap.

We thank the Reviewer for the suggestion, but unfortunately the MATLAB toolbox that we used for plotting the whole-brain effects of the controls (bspmview) doesn't allow us to overlay areas of activations with dashed contour. In the attempt to follow the Reviewer's advice, as a compromise solution we have now added the relevant slices in Supplementary Figure 8 so that readers can compare the results more directly.

10: Most of the analyses are based on correlations. I would be much more convinced by decoding/predictive analyses. Could the authors attempt to decode performance base on activation zone for example?

We are not sure what the Reviewer refers to as “decoding/predictive” analyses, especially in light of the fact that we do report a crucial decoding analysis in the results (Fig. 2f). This analysis was a cross-context decoding, meaning that we trained our classifier on one context and we tested it on the other, thus proving the existence of egocentric representations aligned across environments. Moreover, hexadirectional analyses are very specific and, at the present moment, no decoding approach has been developed to investigate grid-like codes.

11. I am very unconvinced as to the interpretation of the eye tracking experiments. See major concern. Please compare analyses between “real” 2D space and conceptual space. Please justify better why eye tracking can apply to a conceptual space.

We hope we replied to the major concern of the Reviewer in our first point, discussing the contrast between “real” and “conceptual” spaces. As regards the justification of why eye tracking can apply to a conceptual space, our logic was the following: besides the reported patterns of brain activation, is there behavioral evidence that participants are internally representing the relationship between their own current viewpoint/state and the target goal using a first person perspective? This is an empirical question, which builds on a large body of research showing that eye movements reflect relational memory and in general reveals important insights about internal mental processes (e.g., Hannula et al. 2010). For instance, in unidimensional conceptual spaces (e.g. the numerical/arithmetic space) eye movements can reliably predict the relative magnitude of freely generated numbers (Loetscher et al. 2010, CurrBio) and correlate with the arithmetic calculation mentally performed by the subjects (e.g., Knops et al. 2009, Science). Similarly, here they reveal the relative egocentric position of a goal in a 2D conceptual space, building up on and extending relevant findings in related fields. Interestingly, in some of these experiments eye movements have been associated with the activity of both the hippocampal formation (e.g., Hannula & Ranganath 2009, Neuron) and the parietal cortex (e.g., Knops et al. 2009, Science).

Moreover, we would like to highlight a couple of aspects on this analysis:

- firstly, the validation procedure performed with DeepMReye in our study: as explained in the text, we did not analyze immediately our eye-tracking data of interest (imagination period), rather we first validated the ability of the analytical approach to distinguish between left vs right using an independent partition of the data, that is the period of answer choice (Yes vs No, or No vs Yes). Only after having checked for the validity of the approach, we applied the same analysis to the period of interest. This makes this approach robust and reliable, in our opinion.
- secondly, we implemented a control for our main analysis of interest about the left vs right effect, namely we verified that no correlation with eye movements should be seen in the vertical dimension. This control is important, as the stimuli that participants are seeing actually vary along the vertical axis during the morphing period. This analysis demonstrates that eye movements are not correlated with this dimension but rather with the horizontal one, which doesn't really carry information on the "real screen/space" but putatively conveys all the information in the conceptual space. This is of particular relevance in our opinion;
- finally, we showed that there was no difference between the two contexts, with the effect going in the (expected) direction for both: this further supports the idea that the two conceptual environments are potentially realigned to maintain the same egocentric point-of-view to the goal.

Please notice that we also have toned down the interpretation of the eye movement results when discussing them in the revised manuscript.

12. Grid-like signal: it is not clear which structures are analyzed: Only mEC and IEC?

We start our analysis with Regions of Interest in the mEC and IEC, following previous studies in the field, then we moved to the whole-brain level to explore the possibility that other brain regions might show the same effect, as reported in some studies (e.g., Constantinescu et al. 2016) but not in others (e.g., Viganò & Piazza 2020; Viganò et al. 2021) - a difference that, must be said, might also relate to the different grid-analysis employed, namely hexadirectional coding based on quadrature filters vs RSA

Reviewer #3 (Remarks to the Author):

The manuscript by Vigano and colleagues describes a human fMRI study in which subjects learned the locations of 2 different stimuli each in a 2-dimensional conceptual space. Subjects were required to “navigate” to these goals in order to perform a behavioral task. While the 2-d conceptual space was not made explicit to subjects, they nonetheless performed well on the task with their behavioral responses and even their eye movements suggesting that they did represent the 2-dimensional space and did so from an egocentric perspective. fMRI analyses (using adaptation and multivoxel pattern analyses) provided converging support for an egocentric coding of space, particularly in medial parietal cortex, that reflected the relative orientation of subjects as they navigated the space on each trial. Separately, entorhinal cortex (and other areas) showed a grid-like allocentric representation of space (at least based on fMRI adaptation measures) that was relatively stronger when subjects were close to the goal location.

The manuscript addresses a very interesting question about whether humans form and can navigate conceptual spaces in an egocentric reference frame. The analyses are well motivated by prior work and complement prior findings. However, the current findings do represent a significant advance on prior work by considering the egocentric navigation. The experimental paradigm is clever and the results include a commendable number of converging analyses. While I think some of the analyses are a bit opaque/dense, I was able to follow the logic and did not see major problems in their implementation or interpretation. Overall, I found this a very interesting paper and believe it will likely be of broad interest. I only have a few comments/questions.

We thank the Reviewer for the very positive feedback. We provide a point-by-point response below to address his/her comments.

1. Could the egocentric reference frame simply reflect whether subjects were getting closer or farther to the goal? From the analyses reported, it's hard to know whether the egocentric representation was fine-grained (sensitive to minor changes in angle) or very coarse (e.g., getting closer or farther). And with the 45 degree increments, maybe this is not addressable. But do the authors think the representation was more fine-grained than closer/farther? If so, is there any aspect of the data that would support this?

There are two types of distance. Euclidean distance: getting closer or farther to the goal can be represented as a change in the distance from it, but this is controlled by the additional analysis reported in Supplementary Figure 10, showing that our adaptation analysis was not affected by goal distance. There is also “angular” distance to the goal (e.g., 45, 90 or 135°). However, also in this case, our egocentric-like coding does not simply reflect whether subjects were getting closer or farther to the goal, since it distinguishes clearly between condition with the same angular distance (e.g. 45°) but on different “sides” of the observer relative to the goal (-45° vs +45°). This property of the egocentric-like representation can be easily observed in the MDS results figure 2h.

The same figure may help to address, in part, the Reviewer's question about granularity. The MDS suggests that the three angular differences are separable in the

neural space, especially for the right side, in which the points for 45°, 90° and 135° are not overlapping. This is not totally the case for the left side, where 135° and 90° are more overlapping. Also in this case, however, this would be particularly interesting because for solving the task there is no need, in principle, to differentiate between angular differences across off-target trajectories: showing even a coarse distinction is an interesting point that supports our claims.

2. Was the distribution of angles to the target fully independent of the starting position? Of particular concern, was there any correlation such that certain angles tended to be closer/farther from the goal? The distance-controlled analysis suggests this might not be a problem, but it also made me wonder if this was a confound in the design (and that was the motivation for the distance-controlled analysis)?

We thank the Reviewer for the question. The average correlation across trials between egocentric angles to target and Euclidean distance to the target at starting positions was Pearson's $r = 0.03$, thus negligible. We have now added this information in the Methods Section. Nevertheless, the control analyses in Supplementary Figure 10 and Supplementary Figure 12 were motivated to reassure that any minor influence of these factors were not affecting our results.

3. Similarly, it would be informative to know if the parietal regions code the distance to the goal. It would be striking, for example, if the parietal regions do not code distance to goal, but do code for angle. This would make for an even stronger argument about egocentric navigation in conceptual space.

This is an interesting point indeed. As reported in Supplementary Figure 10, there are significant clusters in the parietal cortex for the distance to the goal, although the strongest activity is in the hippocampus. Therefore, some information about distance is definitely present in the parietal cortex, and this is confirmed also by an additional Region of Interest analysis: by using the cluster in the medial occipito-parietal cortex shown in Figure 2c as ROI, we observed that there was moderate but significant modulation of the BOLD signal as the one modeled by the distance regressor (mean parameter estimate: 0.31, $p = .02$), that probably did not survive whole-brain correction. However, this effect was significantly lower than that of egocentric conditions modeled in our main analysis (p value of the difference: .005), and this is consistent with the whole brain results reported in Supplementary Figure 10b, where we show the areas that respond to the egocentric direction after controlling for distance.

In short, in our experiment the parietal cortex shows a weak modulation of distance to the goal, but its activity is mostly representing egocentric direction.

4. Were subjects debriefed in any way to gather information about their strategy for performing the task? Although they were not explicitly made aware of the 2d structure, did some/any of the subjects figure this out on their own?

No systematic or standardized debriefing was conducted, but we asked participants at the end of the experiment if they had used any particular strategy or if they had

represented somehow explicitly the stimuli and their relations. None of them reported the use of any particular strategy rather than evaluating the similarity with the goal.

REVIEWERS' COMMENTS

Reviewer #1 (Remarks to the Author):

The authors have given thoughtful consideration to all my points, and have made appropriate edits which improve the article. The addition of the videos is also very helpful. As I was already impressed with the initial submission, I think the revised version is almost ready for publication.

My only remaining reservation concerns the interpretation of the eye movement data touch on in various responses and especially in the second part of their response to my point 14.

My comment: "I think I am right that the orientation of the conceptual space is arbitrary such that (allowing for the different bond-lengths to be represented on different axes and with arbitrary different signs) each participant might have their own sense of clockwise/anticlockwise, left and right, in the conceptual space? If so this also means that (unless I am missing something) the eye movement effects are difficult to predict or interpret. Perhaps the observed effects are related to biases in the allowed rotation in the collect task or spatial biases (such as reading direction) that participants bring to the task/stimuli?"

Their response: "... a priori, we could not make strong predictions about a specific sense and also for that reason, maybe, the eye movement results should be taken as preliminary - as we now stress in the paper. On the other hand, however, the Reviewer is right in thinking that subjects may arrive with some common bias that could explain a common configuration of the conceptual space that can be reflected in eye movements."

I think the authors accept my argument that the left-right clockwise-anticlockwise sense of the egocentric conceptual space they are probing is arbitrary, and therefore we might expect a 50:50 split among participants. The eye-tracking responses do not reflect this, but I am not clear on the authors explanation: "we consider more likely that the clockwise (counter-clockwise) movements in terms of egocentric bearing from a reference vector (the goal vector) correspond to rightward (leftward) movements as it would naturally happen if the goal vector reflected the subject's facing direction." My point here is that if the conceptual axes were flipped arbitrarily, the same stimulus could correspond equally to a rightward (clockwise) or a leftward (anticlockwise) movement in egocentric space. Either this is i) a stubborn misconception on my part, in which case the authors should patiently explain it in the main text of the article, or else ii) it is correct, in which case the authors should address it explicitly if briefly in the main text (for example, "Although the DeepMReye results tend to support the egocentric spatial interpretation of mental search in our task they also raise a puzzle in that they suggest that the large majority of participants must have adopted the same axis orientation for this space; the axis orientation (e.g., upper bond-length increases to the right in figure 1b) is in fact arbitrary, so the presence of a consistent bias among participants is perhaps surprising.").

Reviewer #3 (Remarks to the Author):

I am satisfied with the revisions that the authors have made. My initial review was quite positive given the very clever design and rigorous analyses. I feel that the revisions (though not extensive) have improved the manuscript.

One very minor type-o:

Line 231: change "complementary" to "complement"

Replies from the Authors are in blue and bold font.

REVIEWER COMMENTS

Reviewer #1 (Remarks to the Author):

The authors have given thoughtful consideration to all my points, and have made appropriate edits which improve the article. The addition of the videos is also very helpful. As I was already impressed with the initial submission, I think the revised version is almost ready for publication.

My only remaining reservation concerns the interpretation of the eye movement data touch on in various responses and especially in the second part of their response to my point 14.

My comment: "I think I am right that the orientation of the conceptual space is arbitrary such that (allowing for the different bond-lengths to be represented on different axes and with arbitrary different signs) each participant might have their own sense of clockwise/anticlockwise, left and right, in the conceptual space? If so this also means that (unless I am missing something) the eye movement effects are difficult to predict or interpret. Perhaps the observed effects are related to biases in the allowed rotation in the collect task or spatial biases (such as reading direction) that participants bring to the task/stimuli?"

Their response: "... a priori, we could not make strong predictions about a specific sense and also for that reason, maybe, the eye movement results should be taken as preliminary - as we now stress in the paper. On the other hand, however, the Reviewer is right in thinking that subjects may arrive with some common bias that could explain a common configuration of the conceptual space that can be reflected in eye movements."

I think the authors accept my argument that the left-right clockwise-anticlockwise sense of the egocentric conceptual space they are probing is arbitrary, and therefore we might expect a 50:50 split among participants. The eye-tracking responses do not reflect this, but I am not clear on the authors explanation: "we consider more likely that the clockwise (counter-clockwise) movements in terms of egocentric bearing from a reference vector (the goal vector) correspond to rightward (leftward) movements as it would naturally happen if the goal vector reflected the subject's facing direction." My point here is that if the conceptual axes were flipped arbitrarily, the same stimulus could correspond equally to a rightward (clockwise) or a leftward (anticlockwise) movement in egocentric space. Either this is i) a stubborn misconception on my part, in which case the authors should patiently explain it in the main text of the article, or else ii) it is correct, in which case the authors should address it explicitly if briefly in the main text (for example, "Although the DeepMR eye

results tend to support the egocentric spatial interpretation of mental search in our task they also raise a puzzle in that they suggest that the large majority of participants must have adopted the same axis orientation for this space; the axis orientation (e.g., upper bond-length increases to the right in figure 1b) is in fact arbitrary, so the presence of a consistent bias among participants is perhaps surprising.”).

We thank the Reviewer for the comments, suggestions, and positive feedback on our work. We are grateful to the Referee and believe that they paid attention to an important aspect of our results. We agree that an explicit addendum in the main text would be useful, therefore we slightly adapted their suggestion in the following way and we have added it to the Discussion section:

“Although the DeepMRye results tend to support the egocentric spatial interpretation of mental search in our task, they also raise a puzzle in that they suggest that the large majority of participants must have adopted the same axis orientation for this space; the axis orientation (e.g., upper bond-length increases to the right in figure 1b) is in fact arbitrary, so the presence of a consistent effect among participants is perhaps surprising, and might reveal the existence of internally biased representations for conceptual spaces similar to those observed in the physical environment (e.g., the left-to-right orientation in enumeration or reading in most of Western cultures)”.

Reviewer #3 (Remarks to the Author):

I am satisfied with the revisions that the authors have made. My initial review was quite positive given the very clever design and rigorous analyses. I feel that the revisions (though not extensive) have improved the manuscript.

One very minor type-o:

Line 231: change “complementary” to “complement”

We thank Reviewer 3 for the positive feedback, we have corrected the typo.